

# The Macquarie Island [LoFlo2G] high-precision continuous atmospheric carbon dioxide record

Ann R. Stavert[1], Rachel M. Law[1], Marcel van der Schoot[1], Ray L. Langenfelds[1], Darren A. Spencer[1], Paul B. Krummel[1], Scott D. Chambers[2], Alistair G. Williams[2], Sylvester Werczynski[2], Roger J. Francey[1], and Russell T. Howden[1]

[1]CSIRO Oceans and Atmosphere, Aspendale, Victoria, 3195, Australia
[2]Australian Nuclear Science and Technology Organisation, Kirawee, New South Wales, 2232, Australia

*Correspondence to:* Ann Stavert (ann.stavert@csiro.au)

**Abstract.** The Southern Ocean (south of 30°S) is a key global scale sink of carbon dioxide ($CO_2$). However, the isolated and inhospitable nature of this environment has restricted the number of oceanic and atmospheric $CO_2$ measurements in this region. This has limited the scientific community's ability to investigate trends and seasonal variability of the sink. Compared to regions further north, the near-absence of terrestrial $CO_2$ exchange and strong large-scale zonal mixing demands unusual

5  inter-site measurement precision to help distinguish the presence of mid-to-high latitude ocean exchange from large $CO_2$ fluxes transported southwards in the atmosphere. Here we describe a continuous, in-situ, ultra-high-precision, Southern Ocean region $CO_2$ record, which ran at Macquarie Island (54°37' S, 158°52'E) from 2005–2016 using a 'LoFlo2' instrument, along with its calibration strategy, uncertainty analysis and baseline filtering procedures. Uncertainty estimates calculated for minute and hourly frequency data range from 0.01 to 0.05 $\mu mol\,mol^{-1}$ depending on averaging period and application. Higher precisions

10  are applicable when comparing MQA LoFlo measurements to those of similar instruments on the same internal laboratory calibration scale and more uncertain values are applicable when comparing to other networks. Baseline selection is designed to remove measurements that are influenced by local, Macquarie Island, $CO_2$ sources, with effective removal achieved using a within-minute $CO_2$ standard deviation metric. Additionally, measurements that are influenced by $CO_2$ fluxes from Australia or other Southern Hemisphere land masses are effectively removed using model-simulated radon concentration. A comparison

15  with flask records of atmospheric $CO_2$ at Macquarie Island highlights the limitation of the flask record (due to corrections for storage time and limited temporal coverage) when compared to the new high-precision, continuous record; the new record shows much less noisy seasonal variations than the flask record. As such this new record is ideal for improving our understanding of the spatial and temporal variability of the Southern Ocean $CO_2$ flux particularly when combined with data from similar instruments at other Southern Hemispheric locations.

## 1 Introduction

Greenhouse gases, such as $CO_2$, released by human activity are primarily responsible for global warming over the last century. Hence, understanding the sources, sinks and feedback mechanisms of these gases is essential for managing the anthropogenic impact on the earth's ecosystems. The Southern Ocean and Antarctic regions, remote from significant industrial and terrestrial





biosphere activity, are ideally located to measure global scale changes and long-term trends in the concentrations of these gases. CSIRO focusses its greenhouse gas sampling program on the Southern Hemisphere, with long-running flask measurements (Francey et al., 1999), currently at 8 sites and in-situ $CO_2$ measurements originally using Non Dispersive InfraRed (NDIR) instruments and now mostly using laser based spectroscopic instruments (4 current long-term sites, 1 shipboard and various

campaigns). With continuing innovation in measurement technology and interpretive models, atmospheric measurements can make a significant contribution to detecting possible climate-induced regional changes in carbon uptake, particularly in the crucial Southern Ocean (SO) $CO_2$ sink, as well as to monitor global changes.

The annual basin-scale Southern Ocean carbon flux is generally well constrained (Lenton et al., 2013). However, the seasonality, long-term trend, interannual and regional variability of this flux is still poorly understood, with divergence between

the ocean biogeochemical models, oceanic inversions, atmospheric inversions and (sparse) observations. Considering that up to a third of the global anthropogenic $CO_2$ uptake by oceans occurs in the SO (the region south of 44°S) (Lenton et al., 2013) accurate quantification of this sink is key. However, efforts have so far been hampered by the limited number of observations currently available (both ocean $pCO_2$ and atmospheric $CO_2$) and their spatial distribution across the Southern Ocean region.

With few representative locations suitable for measuring atmospheric $CO_2$ in the Southern Ocean, Macquarie Island (54°37'

S, 158°52'E) was recognised as a potential monitoring location in the 1970s. The island is ideally situated in the middle of the Southern Ocean near the subantarctic front, the boundary between the subantarctic zone and polar frontal zone. This is a highly active oceanic region, known to be a $CO_2$ sink in the summer months due to biological production, and a $CO_2$ source in some areas during winter as a result of deep water mixing (Lenton et al., 2013).

A key challenge when measuring atmospheric $CO_2$ at Macquarie Island is the limited access. In-situ monitoring of at-

mospheric $CO_2$ was attempted in 1979 but the restricted access to the island limited the supply of calibration and reference gases. This, along with the intermittent operation of the NDIR, contributed to observations of insufficient quality to be scientifically useful. Macquarie Island was included in CSIRO's flask-sampling network in 1986, with data regularly submitted to international archives from 1990. However, long delays between collection and measurement for flask samples at locations resupplied only once per year along with instrument performance at the time limited their accuracy (Cooper et al., 1999; Sturm

et al., 2004). Consequently a new "LoFlo" in-situ $CO_2$ instrument was installed at Macquarie Island in 2005 (LoFlo2G) taking advantage of technological advances to significantly improve instrument performance, cylinder stability and calibration strategies. While the performance of the instrument has been outstanding (see below), uncertainty about future logistical and staffing constraints at Macquarie Island has necessitated replacing the ageing LoFlo in late 2016 with a newer more-linear spectroscopic instrument with precision approaching that of the LoFlo, but having lower maintenance requirements, and im-

proved temporal coverage due to reduced calibration requirements. While it operated the Macquarie Island LoFlo was part of a Southern Hemisphere LoFlo network comprising instruments at Cape Grim, Tasmania (144.7°E, 40.7°S), Amsterdam Island (77.5°E, 38.0°S) and Baring Head, New Zealand (174.9°E, 41.4°S). A further LoFlo instrument (LoFlo2B) based at CSIRO (Aspendale, Australia) is used for calibration and related tasks, as well as occasional monitoring of local air.



This paper focuses on the technical aspects of the Macquarie Island in-situ $CO_2$ measurement program, including site details, instrumentation and calibration (Sections 2 and 3), data characteristics and comparison with the flask record (Section 4). Data selection for 'baseline' conditions is considered in Section 5, and Section 6 gives a general climatology of the $CO_2$ data set.

## 2   Site description

Macquarie Island is 34 km long and 5 km wide at its widest point (Russ and Terauds, 2009) (Fig. 1b). It lies on a north-south axis and has an area of 12788 hectares. Located approximately 1500 km south east of Australia, and 1600 km north of the Antarctic continent, it is ideally situated for Southern Ocean based studies (Fig. 1c). Macquarie Island has a mean minimum and maximum temperature of 3.1 °C and 6.6 °C and average annual rainfall of 981.6 mm (http://www.bom.gov.au/climate/-averages/tables/cw_300004.shtml). Winds are predominantly from the west (35 %), northwest (35 %) and north (15 %), with

an average wind speed greater than 9 ms$^{-1}$. The island is extremely windy with the winds classed as calm less than 1 % of the time.

The Clean Air Laboratory is located on a low-lying (6 m above sea level, a.s.l.) isthmus between the main body of the island (a plateau 200–400 m a.s.l.) and a small hill at the northern end of the island (Fig. 1a). It is ~150 m from the residential section of the station, west ("upwind") of local anthropogenic point sources of $CO_2$ (the incinerator and powerhouses, Fig. 1a). The area surrounding the laboratory is highly biologically active, rich in both aquatic and terrestrial flora and fauna which,

considering the relatively low intake height (13 m a.s.l.), may impact $CO_2$ measurements under low wind speed conditions. A heated concrete floor helps to maintain the laboratory at 19 °C, but on warm sunny days this may drift slowly by 1–2 degrees.

Maintaining an in-situ instrument on Macquarie Island is logistically challenging. Since there is no airport, access has been restricted to an annual resupply voyage in March or April. All instrument servicing must be completed in the "resupply

window", which is generally less than a week. As the resupply ship cannot dock on the island, all equipment and personnel must be transported from the ship to the shore by either helicopter or small boat. These restrictions make Macquarie Island less accessible than many Antarctic sites, possibly the most inaccessible of all sites in the current $CO_2$ monitoring network. Between resupply visits, Bureau of Meteorology observational and technical staff were responsible for flask sampling and general maintenance of the in-situ instrument and drying system. Instrument diagnostics and calibration runs were performed

remotely. All communication with the island is via a restricted satellite link.

The Clean Air Laboratory also houses an atmospheric radon monitor, the output of which can be useful for interpreting the $CO_2$ record. A 700 L dual-flow-loop two-filter radon detector (Whittlestone and Zahorowski, 1998; Chambers et al., 2014) was installed during the 2011 re-supply visit. The detector samples ambient air from an inlet approximately 5 m a.g.l., at a flow rate of ~45 Lmin$^{-1}$. A 400 L delay volume was incorporated within the inlet line to allow for the decay of the short-lived radon

isotope thoron ($^{220}$Rn, $T_{1/2}$=56 s). The detector has a response time of around 45 minutes, and a lower-limit-of-determination (defined here as the radon concentration at which the detector's counting error is 30 %) of ~40 mBqm$^{-3}$. During routine operation the detector is calibrated monthly, by injecting radon from a well-characterised Pylon Radium-222 source ($^{226}Ra$, 19.58 kBq ± 4 %) for 6 hours at a low rate of ~170 ccmin$^{-1}$, and instrumental background checks are performed quarterly.





Problems that arose with the calibration unit and sampling stack-blower, which were not able to be addressed until subsequent re-supply visits, have limited the data availability and accuracy of the absolute calibration until April 2013.

## 3   Experimental design and instrumentation

### 3.1   Continuous $CO_2$ Instrumentation

Carbon dioxide mole fractions have been measured from April 2005 until October 2016 using a CSIRO LoFlo Mark2 $CO_2$ analyser. This analyser is an integrated system constructed around a Li-COR (LI-6262, Li-COR Inc., Nebraska, USA) Non Dispersive Infrared (NDIR) optical bench. The early design of this system is described in Da Costa and Steele (1999) while details of subsequent calibration strategy and software control development are documented in Francey et al. (2004). The internal Li-COR analyser is operated in differential mode where the raw measurement signal is reported twice a second as the

difference in $CO_2$ mole fraction between the sample and reference cells rather than an absolute measurement of $CO_2$. This has the great advantage that the effect of any enviromental variables affecting both cells (e.g. temperature) cancel. The inclusion of tight control on the differential pressure, temperature and flow rate (requiring additional unconventional feedback circuitry to avoid polymer surfaces contacting with the measured air stream) underpin improved precision over conventional NDIR.

Dual stage regulators (high purity, stainless steel, 64-3400 series, Tescom Corporation, Elk River, Minnesota, USA) are used

on all reference and calibration cylinders, and all fittings and tubing used throughout the system are stainless steel. Each hour the instrument alternates between 10 minutes of reference measurement (when reference gas is passed through both cells of the Li-COR) and 50 minutes of sample measurement (reference in one cell and sample gas in the other). While temperature, pressure and flow rates are tightly controlled within the system, small variations in flow and pressure occur following the switch between sample and reference modes. Consequently, the first 6 minutes after a switch are excluded to ensure that the flow and

pressure have stabilised. The performance of the instrument over the remaining 44 minutes is explored further in Section 3.4.2. Short-term (between hour) instrumental drift is removed by deducting the mean raw value of the bracketing reference gas measurements from the sample measurement. Cylinders of dry Southern Ocean air, collected during baseline periods (winds S to SW, wind speed > 5 $ms^{-1}$) at Cape Schanck, Australia (38°29'S, 144°53' E), are used as the reference gas to reduce matrix matching effects between the reference gas and sample air (LI-COR Inc., 1996). The reference gas is stored in 29.5

L high-pressure aluminium cylinders (Luxfer Gas Cylinders, Riverside, California, USA) with each cylinder being used for approximately 6 months.

Despite the remote location of the instrument, instrument performance has been remarkable with only 3.4 % of collected data points rejected due to poor instrumental performance (software failures and sporadic flow rate and temperature issues). Many of these were in the first year, with the annual average data lost for 2006 onwards being only 2.3 %.



## 3.2 Continuous $CO_2$ intake, drying system and servicing

Ambient air is sampled from 7 m above ground level (13 m a.s.l.) through an inverted stainless steel cup with a 4 mm mesh covering the inlet. Quarter inch polymer coated aluminium tubing (Dekoron® "1300") is used between the inlet and pump manifold with the intake line positioned so a continous descent towards the pump is maintained. A simple manifold system,

consisting of 2 and 7 μm filters (SWAGELOK, FW series), pressure gauge (Swagelok PGI-63C-PG15-LAOX 15psig), back pressure regulator (0–15psi ITT Conoflow GH30XTHMXXXB) and flow meter (Dwyer VFA-24-SS 10 L min$^{-1}$) is used. Air is drawn through this manifold using a KNF pump (KNF PM 17835-86 with stainless steel head, PTFE-coated viton diaphragm and PTFE valve plate) at a rate of 5 to 7 L min$^{-1}$. A small volume of air (~30 ml min$^{-1}$) is split from the main flow before the back pressure regulator and enters the drying system. The back pressure regulator, set to between 6 and 7 psig, is used to

control this flow.

Air entering the drying system is immediately split in two: half is dried using two 200 mL drying towers filled with Magnesium percholorate, the other half, the air entering the LoFlo, is dried using a Nafion drier. To minimise $CO_2$ exchange across the Nafion membrane the chemically dried air is used as the 'dry' air stream of the Nafion drier. This prevents a $[CO_2]$ gradient forming between the dry and wet air streams of the Nafion.

Internal drying reagent and $CO_2$-absorbing reagent in the Li-COR system along with the 2 and 7 μm filters and pump diaphragm and valve plate are replaced annually. The $1/4$" tubing between the cup and the pump manifold was replaced in April 2010 and the intake cup cleaned.

## 3.3 Continuous $CO_2$ calibration

Macquarie Island LoFlo measurements are made relative to an assigned concentration of the reference cylinder consisting of

Southern Ocean ambient air (see Section 3.1) minimising the impact of the instrument's nonlinear response and the influence of surface memory effects which occur when switching between reference and sample measurements. The concentration of this reference cylinder is assigned during calibration runs, conducted every 4–6 weeks, as previously described by Steele et al. (2003) with a repeatability of 0.004 μmol mol$^{-1}$ over the average lifetime of the reference cylinder (See Section 3.4.1 for further details). These runs are made using a suite of cylinders with mole fractions spanning the range of concentrations

typically observed at MQA. These cylinders remain permanently attached to the LoFlo, via stainless steel tubing, minimising delays due to surface equilibrium and any risk of contamination. At current calibration gas consumption rates a calibration suite is expected to have a lifetime (around 40 years) significantly greater than that of the instrument.

Calibration runs consist of alternating 5 minute reference (reference in both cells) and calibration (reference in one cell and calibration gas in the other) measurements. As for the normal sampling measurement procedure, the bracketing reference

measurements are deducted from the calibration gas measurement to remove short-term instrumental drift. During a calibration run the cylinders are measured first in ascending, and then in descending order of $CO_2$ mole fraction. Eighteen such "calibration pyramids" are collected during each calibration run. A full calibration run with 7 cylinders takes 5040 minutes (3.5 days). For



each calibration run the response function of the LoFlo system (a shallow quadratic), and the $CO_2$ mole fraction of the reference gas, are determined.

Like the reference gas cylinders, calibration cylinders are made using dry Southern Ocean air collected at Cape Schanck, which is then modified to achieve targeted mole fractions higher or lower than ambient using aliquots of pure $CO_2$ or "$CO_2$ free air" (air which has had the $CO_2$ chemically stripped during collection). The concentrations of these MQA calibration cylinders are made using the Aspendale LoFlo (LoFlo 2B) following an identical procedure to that described above. The concentrations of the LoFlo 2B suite have been provided by the WMO Central Calibration Laboratory (CCL) made using conventional NDIR relative to the WMO X2007 scale (Zhao and Tans, 2006). This calibration propagation pathway is shown in Fig. 2e. Mole fraction assignment through the LoFlo instrument (typical uncertainty <0.01 μmol mol$^{-1}$, Francey et al., 2010, supplementary material) has been shown to be more precise than that of conventional NDIR (0.07 μmol mol$^{-1}$, Zhao and Tans, 2006).

The LoFlo2B calibration suite was calibrated directly against the WMO X2007 scale by the CCL on two occasions, 8 years apart. Differences for individual cylinders varied, averaging 0.01 μmol mol$^{-1}$ over the 8 year period. As these differences do not vary consistently with time or concentration it is likely that these differences reflect random uncertainty in the CCL's measurement method rather than actual changes in $CO_2$ mole fraction. As such, $CO_2$ assignments used here are the mean values of the two CCL calibrations. A detailed uncertainty analysis of this calibration approach is given in Section 3.4.

Two $CO_2$ calibration suites, each containing seven 29.5 L high-pressure aluminium cylinders, have been used at Macquarie Island. The first suite, Suite 2G-a (Table 1) was installed with the system in 2005. However, this suite was accidentally partly vented, and was replaced in April 2006 with a second calibration suite, Suite 2G-b (Table 1). In March 2009 it was decided to stop using the lowest $CO_2$ cylinder of Suite 2G-b as its mole fraction (317.64 μmol mol$^{-1}$) was far lower than mole fractions observed at MQA. For comparison the two LoFlo2G suites (2G-a and 2G-b) and reference cylinders were also measured using gas chromatography (Francey et al., 2003) giving very similar mole fraction to those determined using LoFlo2B (Table 1).

## 3.4 Error propagation

It is important to characterise measurement uncertainty given the small atmospheric signals at mid-high latitudes in the Southern Hemisphere. An earlier study documents the impact of measurement errors and biases of LoFlo, conventional NDIR and flask measurements on $CO_2$ growth rate estimation at Cape Grim, a key Southern Hemisphere site (Francey et al., 2010). Here, our approach is to quantify the measurement uncertainty of the MQA $CO_2$ observations by examining each of five possible sources of error and how they contribute to the uncertainty of hourly and minutely mean values.

MQA measurements were calibrated following a multi-stage protocol (Fig. 2e) which uses a shallow quadratic nonlinearity correction, based on the difference between the reference and sample raw instrumental response and the fixed mole fractions of the calibration standards (Section 3.3). Key sources of uncertainty in this approach, are:

1. The random uncertainty in measuring the $CO_2$ difference between two gases (Type 1)

2. The accuracy of the nonlinearity correction with changes in the absolute mole fraction difference between the reference and sample at both the minutely and weekly timescale (Type 2)





3. Systematic within-hour variation in the sample-reference $CO_2$ difference during the 50 minute sample measurement period (Type 3)

4. The mole fraction stability of the reference standard over time (Type 4)

5. The propagation of mole fractions to the 2G calibration suites from the WMO X2007 scale via the LoFlo2B instrument (Type 5)

Here we quantify each of these five contributions to measurement uncertainty, thus providing a framework for defining uncertainties specific to data applications, e.g. involving different averaging periods or comparison with other data sets. Combining uncertainties of all five types in quadrature defines the overall measurement uncertainty when comparing measurements, including those of other laboratories, that are independently calibrated against the WMO x2007 scale. Comparisons of measurements made within the CSIRO network on similar instruments relative to LoFlo2B will have significantly smaller Type 5 component. The uncertainty analysis uses only data with stable instrumental temperature and pressure and also excludes measurements made shortly after valve switches to minimise line conditioning effects. Uncertainties inherent in the sample handling or intake system, involving potential modification of sample air before being admitted to the LoFlo instrument, have not been examined.

### 3.4.1 Type 1 and Type 2 uncertainty: the random uncertainty in measuring the $CO_2$ difference between two gases and the accuracy of the nonlinearity correction with changes in the absolute mole fraction difference between reference and sample

These two uncertainty types were assessed using regular measurements of the second suite of calibration standards (2G-b) as a proxy for in-situ air data. This analysis was based on eighty calibration runs between 2006 and 2013. Each calibration run included between 16 and 144 (mean = 84) minutes of retained raw data for each individual calibration standard.

Minute-mean mole fractions of the calibration standard data (i.e. the proxy air samples) were calculated for each run using the nonlinearity correction determined in the previous calibration run. This represents a worst-case scenario, as in-situ mole fractions will generally be calculated using a nonlinearity correction determined much closer in time and will not be affected by any regulator or gas handling/switching effects.

First we examined uncertainty in the nonlinearity correction characteristic of the one minute timescale. The minute-mean 1-sigma uncertainties of these proxy air samples were determined, for each calibration standard, as the mean 1 minute standard deviation for each run averaged over the 80 calibration runs. These 1-sigma uncertainties were compared to the absolute mole fraction difference between calibration and reference standards (Fig. 2a). This shows a clear mole fraction dependence, with the 1-sigma uncertainty for a minute mean increasing from $0.025 \, \mu mol \, mol^{-1}$, at close to the reference mixing ratio (this is the Type 1 random uncertainty component inherent in measuring the $CO_2$ difference between two cases due to instrument precision and counting time), to $0.034 \, \mu mol \, mol^{-1}$ when the absolute sample reference mole fraction difference was $70 \, \mu mol \, mol^{-1}$.





The slope of the line is 0.0001, indicating uncertainty of 0.01 % of the sample-reference mole fraction difference at a 1 minute time scale. This Type 2 mole fraction dependent component of uncertainty is negligible for the vast majority of in-situ measurements since at MQA 99.9 % of minute measurements are within 10 $\mu$mol mol$^{-1}$ of the reference standard.

The same data set was used to evaluate uncertainty in the nonlinearity correction over timescales of a few weeks, which
relates to the time period between calibration runs. For this case we calculated the mean $CO_2$ mole fraction per calibration standard per run, still using the nonlinearity corrections defined by the previous calibration runs made typically four to six weeks prior. Variability in the mean from run to run (as expressed by the standard deviation of residuals of these means from the mean of all runs), was plotted against the absolute difference from the reference standard mole fraction (Fig. 2b open circles, a linear fit to the data is shown as the dashed black line). Retained data included 18 runs for standard 994235 and 37 runs for the
other six standards. As such, 18 of the nonlinearity corrections included in this analysis were based on 7 calibration cylinders while the remaining 37 used only 6 cylinders. In this analysis it was assumed that the calibration standard mole fractions were stable, and hence any mole fraction variability was due to changes in the instrumental response.

Standard 994235 was a clear outlier in this analysis (low open circle Fig. 2b). This was the standard dropped from analysis in March 2009 (Section 3.3). This is possibly attributable to a shorter analysis period; less than three years compared to greater
than six years for the other six standards. To investigate this further the analysis was repeated using only runs that included 994235 (18 runs of all 7 cylinders, Fig. 2b small closed circles) and a linear fit to those data (fit a). A linear fit (fit b) to the data from all runs but excluding the standard 994235 data point was also calculated (Fig. 2b black solid line). The slope for fit (a) is shallower than that for fit (b), 0.0003 compared with 0.0008, indicating less uncertainty in mole fractions assigned using calibration runs which included cylinder 994235. This may be due to the tighter constraint on the quadratic fit (i.e. using seven
rather than six calibration cylinders) or possibly a deterioration in instrumental stability over time. There is also evidence of higher variability in instrument non-linearity over longer timescales (weeks vs. minutes), with an eight-fold larger uncertainty found for the ~monthly (Fig. 2b slope 0.0008) compared to the minutely (Fig. 2a slope 0.0001) timeframe.

Interestingly the y-intercept shows the run-to-run random uncertainty for repeat cylinder measurements as 0.004 $\mu$mol mol$^{-1}$. This is independent of the inclusion of cylinder 994235 but is slightly larger than the random uncertainty determined when
RMS scaling the minute-mean Type 1 uncertainty to a matching run length (i.e. $0.025/\sqrt{84} = 0.0027$ $\mu$mol mol$^{-1}$ where 84 is the average number of minutes in a calibration run). This is probably driven by drifts in the calibration cylinder mole fractions over time.

As for the Type 2 uncertainty in minute means, this component is again typically very small, less than 0.008 $\mu$mol mol$^{-1}$ for sample-reference differences of less than 10 $\mu$mol mol$^{-1}$.

### 3.4.2  Type 3 uncertainty: within-hour variation in the sample-reference $CO_2$ difference

Between calibration runs that are performed several weeks apart, the instrument operates in routine in-situ monitoring mode. This involves an hourly cycle of alternating measurement of reference and ambient MQA air. The first 10 minutes of each hour are used for reference measurement (reference in both cells) to determine the difference in output between cells. This difference is used by the data processing algorithm to define a background signal, interpolated between successive reference





measurements made every hour, against which ambient $CO_2$ measurements are subsequently quantified. Ambient air is then admitted to the sample side cell and measured relative to the reference (in the reference side cell) for the remaining 50 minutes of the hour.

The first six minutes of data from both the reference and ambient air measurement periods are excluded from further pro-
cessing due to stabilisation of flow rate and pressure in the sample side cell after the valve switch. For ambient air, $CO_2$ measurements are obtained for the remaining 44 minutes of the hour. However, further investigation into the stability of these data has revealed subtle, systematic drifts in minute-mean $CO_2$ over the 44-minute period.

In order to resolve these small instrumental artefacts in ambient $CO_2$ data, we consider only hours with small atmospheric $CO_2$ variability. Figure 2c shows minute-mean mole fraction deviations from the average of the last 5 minutes in each hour,
averaged by calendar year, and over hours that (i) contain the complete 44 minutes of retained data, and (ii) have a minute-mean standard deviation of $CO_2 < 0.15 \, \mu mol \, mol^{-1}$. For comparison purposes, data are also presented for one year (2011) with no selection for low $CO_2$ variability. This curve is slightly noisier, however the magnitude and time dependence of $CO_2$ deviations is similar to the case with data selection.

The curves for different years are very similar in shape, with deviations being largest in the early minutes and then decaying
to zero at around minute 45. There is a suggestion that the magnitude of deviation has increased over time, with 2006 showing the smallest deviation at minute 16 of $0.02 \, \mu mol \, mol^{-1}$ and 2014 the largest of $0.06 \, \mu mol \, mol^{-1}$. The cause of this within-hour drift has not been confirmed, but is suspected to result from re-equilibration of the internal surfaces of the Nafion drier (Naudy et al., 2014) to disruption of sample air flow during the ten minute reference measurement period.

We assume here that the latter, more stable part of the ambient measurement period provides the most reliable $CO_2$ measure-
ments, and thus construct our hourly dataset using the mean of 30 minutes of data collected between minutes 30–59 of each hour, with a timestamp of 45 minutes past the hour. This is a compromise between maximising the number of minutes contributing to hourly means and limiting any systematic bias associated with the time-dependent drift. The bias in hourly means calculated this way, relative to the last 5 minutes of the hour, is within $0.003 \, \mu mol \, mol^{-1}$. We take this figure to represent the uncertainty characteristic of the within-hour (Type 3) drift that is applicable to the comparison of hourly means.

Definition of the Type 3 uncertainty applicable to minute means is more complex, as it comprises both random and systematic components, varies with minute number within the hour, and in some respects increases with time (i.e. increasing maximum deviation between 2006 and 2014 as displayed in Fig. 2c). For the purpose of quantifying the random component in a way that can be simply integrated with the overall uncertainty analysis presented here, we conducted a second analysis calculating the variability in minute-mean deviation from the mean of minutes 55–59 across all low $CO_2$ variability hours in 2011. This
indicates variability is largest at minute 16 and diminishes to zero by the latter part of the hour, consistent with the earlier description of the magnitude of the artefact. We use the minute 16 figure of $0.02 \, \mu mol \, mol^{-1}$ as a representative estimate of the random uncertainty component. We do not include the systematic uncertainty in subsequent calculations but note that (i) this should be considered in any comparisons of minute mean data and (ii) there is potential to correct for this artefact, for example using the averaged annual behaviour from Fig. 2c.





### 3.4.3 Type 4 uncertainty: stability of the reference cylinders over time

The uncertainty inherent in assuming that the $CO_2$ mole fraction of the reference standard (reference mole fraction) is constant over time was investigated by calculating the change in assigned reference mixing ratio determined from the first and subsequent calibration runs for each of the 18 reference standards. Although the number of calibration runs varied for each

standard, all were analysed at least 3 times (average of 5.7) over a period of 40 to 202 (average of 158) days (Fig. 2d). The mean systematic drift was determined from a quadratic fit to the difference data (black line Fig. 2d), indicating a drift of 0.0017 $\mu mol\,mol^{-1}$ averaged over a month (the average time between calibration runs).

The short-term variability of each cylinder (Fig. 2e) was separated from the systematic drift by fitting, and then subtracting, a quadratic (representing long-term drift) from each standard's set of differences. The standard deviation of short-term variability

values for each standard was determined and the average of all cylinders calculated to give a mean 1-sigma uncertainty of 0.0021 $\mu mol\,mol^{-1}$. Combining the short-term variability and systematic drift results in an overall Type 4 uncertainty of 0.0038 $\mu mol\,mol^{-1}$ in the stability of the reference standard mole fraction.

### 3.4.4 Type 5 uncertainty: propagation of the WMO X2007 scale to the 2G calibration suite

The mole fractions of the 2G calibration suite were linked to the WMO X2007 scale using measurements made on LoFlo2B

against the 2B calibration suite, which is, in turn, linked to the WMO X2007 scale (Fig. 2f). Hence the propagation uncertainty for the 2G calibration suite will consist of both the propagation uncertainty between it and the primary WMO X2007 scale (via the 2B calibration suite) and the uncertainty inherent in 2B measurements. Zhao and Tans (2006) give the random uncertainty associated with propagation of the NOAA primary scale to individual standards as 0.07 $\mu mol\,mol^{-1}$. As such the propagation uncertainty for the 7-cylinder LoFlo2B suite will be 0.026 (i.e. $0.07/\sqrt{7}$) $\mu mol\,mol^{-1}$.

Similarly to the earlier discussion for LoFlo2G the remaining LoFlo2B uncertainties can be separated into Types 1, 2, 3 and 4. Combining in quadrature the 2B propagation uncertainty with Type 1, 2, 3 and 4 uncertainties estimated based on the worst-case 2G uncertainties, the 2G WMO X2007 propagation error was estimated as 0.024 $\mu mol\,mol^{-1}$. This estimate is based on an average run length of 84 minutes of raw data and mean reference-to-sample mole fraction difference of 30 $\mu mol\,mol^{-1}$. This is expected to be an overestimate for the instrumental uncertainties in the 2B data due to the vastly differing laboratory

environments and hence condition of the two instruments. LoFlo2G was developed in the same laboratory as LoFlo2B but has since been transported by sea to MQA, had only limited maintenance (Section 2) and measured predominantly wet, salty ambient air.

### 3.4.5 Overall uncertainty

By geometrically combining appropriate uncertainty types and selecting key factors, it is possible to give a series of examples

of the expected minute mean and hourly uncertainties for different situations (Table 2). These examples all use the worst case Type 3 and 4 uncertainty estimates.



Typically the uncertainty is dominated by the Type 5 uncertainty component, which in turn is comprised mainly of the propagation uncertainty to the WMO X2007 scale. As such, the applicable uncertainty is highly dependent on the network choice, decreasing by up to 40 % when considering within network $CO_2$ comparisons for CSIRO high-precision instruments referenced to the LoFlo2B calibration suite (e.g. the Cape Grim and MQA LoFlos), as compared to between network compar-

isons calibrated to the WMO X2007 scale. For a 30-minute mean observation with a mole fraction near the reference cylinder mole fraction (> 99.9 % of MQA observations) these uncertainties would be, 0.025 µmol mol$^{-1}$ and 0.036 µmol mol$^{-1}$ for within and between network comparisons respectively. In comparison, the increase with sample to reference difference is typically much smaller, for example a 0.003 µmol mol$^{-1}$ increase in uncertainly for a 20 µmol mol$^{-1}$ increase in the sample to reference difference of a 30-minute mean.

## 4   Data characteristics and comparison with flask measurements

### 4.1   Typical features of the $CO_2$ record

Macquarie Island $CO_2$ data display a number of characteristics which we illustrate here by showing a 30 day subset (August 18–September 17 2011) of minute-mean and standard deviation of $CO_2$ mole fractions (Fig. 3). The minute means and standard deviations are calculated from the raw 2 Hz data. The period was chosen because it has good data coverage of both $CO_2$ and

wind data and radon was being measured through this period, although with poor data quality as noted in Section 2.

The minute-mean $CO_2$ mole fractions, shown in Fig. 3, are around 389 µmol mol$^{-1}$, increasing slightly over the 30 days. For most of the period, mole fractions within an hour vary by 0.1–0.3 µmol mol$^{-1}$, while variations over a day are typically 0.5–1.0 µmol mol$^{-1}$. There are also larger positive and negative deviations of 2–3 µmol mol$^{-1}$. The positive deviations (e.g. day 238, day 242 and day 256) are characterised by elevated standard deviation while the negative deviations are not (e.g. day

235, day 247). During this period, flask samples were filled on August 18 (day 230) and September 6 (day 249). The flask mole fractions at day 230 agree reasonably well with the in-situ measurements. By contrast the flasks filled at day 249 do not have good flask pair agreement, with the higher mole fraction flask being around 1.7 µmol mol$^{-1}$ above the coincident in-situ measurement. It is worth noting that this flask had already been flagged as an outlier by the standard flask-fitting and quality checks applied to the flask record. Flask and in-situ measurements are compared across the full in-situ record in Section 4.3.

Figure 4 provides a closer look at one positive deviation and one negative deviation. The increased mole fractions around day 237.8 to 238.0 (Fig. 4b) are at times of lower wind speed (Fig. 4a), indicative of a local influence on observed $CO_2$ mole fractions. In general, as with this example, deviations associated with low wind speed are more often positive than negative, suggesting a contribution from anthropogenic sources as well as biospheric sources and sinks. The categorisation of the minute means by standard deviation (indicated by the dot colour in Fig. 4) shows that large deviations are mostly, but not always,

associated with high standard deviation. This is important to note when considering whether $CO_2$ standard deviation is helpful for data selection (Section 5).

Figure 4e focuses on a negative $CO_2$ deviation around day 235. This deviation is coincident with a change in wind direction from westerly to north-easterly (Fig. 4d) and increased radon concentrations (Fig. 4c), both modelled (see Section 5.2) and





observed. The modelled radon shows a somewhat broader peak than observed but captures the main features of the event. The wind speed through this period (not shown) was greater than $10\,\mathrm{m\,s^{-1}}$. Elevated radon is a good marker of air that has had significant contact with land surfaces over the previous week or so. Consequently, the negative deviation in $CO_2$ mole fraction is likely due to biospheric uptake of $CO_2$. Back-trajectories (not shown) suggest the uptake occurred over Tasmania

and Southern Australia, before the air mass was transported to Macquarie Island. $CO_2$ standard deviations are low throughout this period, with only occasional minutes in the $0.10$–$0.15\,\mathrm{\mu mol\,mol^{-1}}$ range and most of those less than $0.12\,\mathrm{\mu mol\,mol^{-1}}$.

Finally we examine a period without large deviations (Fig. 5). This period shows some sensitivity to wind speed, with more scatter in the minute-mean $CO_2$ around the start of day 256 when the wind speeds are lower, and the coincidence of the high minute $CO_2$ values around day 258.4 with slightly reduced winds. As seen in Fig. 4a, this period also shows occurrences of

higher $CO_2$ standard deviation when the wind speed is lower. Also evident here is what appears to be a diurnal cycle, with lower values around 0–2 UT (11–13 LT). This is more evident in the last two days shown (peak-to-trough amplitude of ~0.5 $\mathrm{\mu mol\,mol^{-1}}$), than the first two days.

In the remainder of this section, and in Section 5, we further explore each of the features identified here, examining how widespread they are across the whole record and the implications for selection of the data record for different purposes.

## 4.2   $CO_2$ standard deviation and wind speed

The distribution of minute standard deviations of $CO_2$ mole fraction for all available data in 2011 is shown in Fig. 6a; other years were similar. The distribution has a mean of $0.076\,\mathrm{\mu mol\,mol^{-1}}$ with a slightly smaller mode (peak), $0.060$–$0.065$ $\mathrm{\mu mol\,mol^{-1}}$. The distribution has a long upper tail, with 1.26% of values between $0.20$–$0.40\,\mathrm{\mu mol\,mol^{-1}}$, and 0.38% above $0.40\,\mathrm{\mu mol\,mol^{-1}}$ (up to the maximum standard deviation of $2.20\,\mathrm{\mu mol\,mol^{-1}}$). The Macquarie Island distribution is com-

pared with the corresponding distribution for 2011 measurements at Cape Grim, Tasmania (144.7°E,40.7°S), made using a similar instrument. Cape Grim standard deviations were generally smaller than for Macquarie Island, with a mean of 0.063 $\mathrm{\mu mol\,mol^{-1}}$ and mode of $0.040$–$0.045\,\mathrm{\mu mol\,mol^{-1}}$. The difference is most likely due to the sampling height and inlet length at the two sites. Cape Grim air is sampled at 70 m from a tower that is on the top of an approximately 100 m cliff. By contrast, Macquarie Island air is sampled from 7 m (13 m a.s.l.).

Figures 4 and 5 suggested a relationship between $CO_2$ standard deviation and wind speed. This can be seen more clearly in Fig. 6b, which shows the distribution of the nearest hourly wind speed to each available minute in 2011 for different $CO_2$ standard deviation ranges. For standard deviations less than $0.10\,\mathrm{\mu mol\,mol^{-1}}$ (almost 90% of all data), the distribution is broad with a peak around $13\,\mathrm{m\,s^{-1}}$. The distribution for the $0.10$–$0.12\,\mathrm{\mu mol\,mol^{-1}}$ standard deviation range is similar with a small increase in the proportion of minutes with wind speed less than $7\,\mathrm{m\,s^{-1}}$. By contrast the distributions for larger standard

deviations are shifted to lower wind speeds, with the peaks of the distribution around 5 and $3\,\mathrm{m\,s^{-1}}$, respectively, for standard deviations between $0.12$–$0.15\,\mathrm{\mu mol\,mol^{-1}}$ and greater than $0.15\,\mathrm{\mu mol\,mol^{-1}}$. For the largest standard deviation category, 87% of the distribution is below $8\,\mathrm{m\,s^{-1}}$. This confirms the hypothesis from the example case above, that $CO_2$ measurements are noisier at lower wind speeds, indicative of an influence from local $CO_2$ fluxes and likely exacerbated by the relatively low





sampling height. The figure also provides evidence that $CO_2$ minute standard deviations may provide a good alternative to wind speed as a criterion for removing local influences from the $CO_2$ record.

### 4.3 Comparison of flask and in-situ measurements

Since 1992, pairs of air samples have been collected fortnightly at MQA, in 0.5 L glass flasks using flask sampling techniques
described by Francey et al. (1996). From 1992–1995 these flasks were sealed with polytetrafluoroethylene (PTFE) O-rings, but since 1996 perfluoroalkoxy (PFA) O-rings were used. Flask sampling is performed when wind speeds are $> 7$ m s$^{-1}$ and the wind direction is from the north-west ($290°$–$360°$) or south-east ($110°$–$180°$) quadrants, to avoid local biogenic and anthropogenic sources and sinks (Fig. 1). Although mounted on the same mast as the LoFlo intake line, the flask sampling intake line, along with its drying and pump systems, are entirely separate to that of the LoFlo.

Filled flasks are stored, and then shipped back annually to CSIRO GASLAB (Aspendale, Australia), where they are analysed for $CO_2$ and its isotopes $\delta^{13}C$ and $\delta^{18}O$, $CH_4$, $H_2$, CO and $N_2O$ (Francey et al., 2003). Data are flagged if the sampled airmass was not representative of baseline conditions, if they were affected by sampling or analytical artefacts, or if they lie more than 3 standard deviations from the 'smoothed curve' fit to the atmospheric record using the methods of Thoning et al. (1989). Flagged data were not used for this analysis.

All measurements derived from CSIRO flask samples require a correction for loss of $CO_2$ with storage time due to perme- ation of gases through the O-rings (Langenfelds et al., 2002; Sturm et al., 2004). These corrections are especially significant for CSIRO's low volume (0.5 L) flasks, and at sites such as MQA where storage times can exceed a year. Loss rates have been determined by comparing data from CSIRO's southern high latitude sites, where flasks can be stored for a year or so before analysis, with smoothed baseline concentrations at Cape Grim, Tasmania, derived from flask sample data with relatively
short storage times. Using data from 1992–2007, a correction of 0.002 µmol mol$^{-1}$ day$^{-1}$ was estimated for flasks fitted with PFA O-rings and filled to 85 kPa above ambient pressure (Langenfelds et al., 2011), leading to storage corrections of up to 1 µmol mol$^{-1}$ for MQA flask samples (Fig 7a, b).

LoFlo observations were compared to individual flask sample data by taking the mean of the hours before and after the flask filling time, or either hour if only one was available. This identified 361 matching records after flagged flasks had been excluded.
Flask-LoFlo concentration differences are shown in Fig. 7c, with differences ranging from -1.3 to 0.9 µmol mol$^{-1}$. Flasks are filled in pairs to help assess measurement quality with the expectation that the two flasks will give similar concentration measurements. At 'clean air' sites such as MQA, flask pair differences are typically expected to be within about 0.1 µmol mol$^{-1}$ for short storage times. Figure 7c shows that when flask pair differences are larger (greater than 0.4 µmol mol$^{-1}$), one of the pair often has an outlying flask-LoFlo difference, suggesting a less reliable flask measurement. There are also cases where there
is only a single flask matched to a LoFlo measurement, and some of these cases also give outlying flask-LoFlo differences. The mean flask-LoFlo mole fraction difference is -0.13 µmol mol$^{-1}$ with a standard deviation of 0.27 µmol mol$^{-1}$. The limitations of the flask record compared to that of the LoFlo instrument are further explored in Section 6 when defining a $CO_2$ climatology for Macquarie Island.



## 5 Defining a baseline record

The aim of most long-term atmospheric $CO_2$ measuring sites is to provide regional 'baseline' $CO_2$ observations. Thus, most sites employ some site specific criteria to select these observations that are considered to be independent of local and point sources and sinks. For flask samples, this selection is largely independent of measurement and often based on some speci-

fied meteorological conditions such as wind speed and direction. For in-situ measurements, selection is a post-measurement process, opening a range of possibilities for different data selection for different purposes. Methods of data selection include meteorological (usually wind) criteria, the concentration of other key atmospheric components (e.g. Rn, Chambers et al., 2016), back trajectories, air mass origin maps, various statistical methods (e.g. El Yazidi et al., 2018) and, due to the high temporal frequency of the measurements, removal of outliers using a statistical fitting procedure (e.g. Thoning et al., 1989). The re-

moteness of Macquarie Island makes defining the baseline record simpler than for many other sites. The aim is, firstly, to remove measurements that are influenced by any local fluxes from the island itself (likely to be small as the land fetch from the predominant wind directions is < 100 m) and, secondly, depending on the application, to remove air samples that have had relatively recent contact with other Southern Hemisphere land (for example Fig. 4e). The selection is applied to the hourly measurement record, noting that hourly-reported mole fractions are actually 30-minute averages, as described in Section 3.4.2.

### 5.1 Removing local flux influences

Local flux influences on the $CO_2$ record are often removed using a wind-speed criterion . Given the relationship described in Section 4.2 between $CO_2$ standard deviation (SD) and wind-speed, here we explore the effectiveness of $CO_2$ SD as a baseline selection method. An obvious advantage of this approach is that it is not dependent on a separate meteorological dataset that may have measurement gaps. A number of $CO_2$ SD measures could be used for this purpose. Based on the behaviour seen

in Fig. 4b, we use the maximum minute $CO_2$ SD contributing to the 30-minute average, i.e. we reject a 30-minute average measurement based on the magnitude of the noisiest minute contributing to that average. Fig. 4b showed that some outlier minute $CO_2$ mole fractions could be associated with relatively low minute $CO_2$ SDs but typically nearby minutes had high minute $CO_2$ SD. Using the maximum minute $CO_2$ SD across the averaging period helps to ensure that any outliers with low $CO_2$ SD are also excluded. We also exclude any 30-minute average which had missing minutes within the averaging period.

The effectiveness of this selection technique has been assessed for a range of "maximum minute $CO_2$ SD rejection thresholds" , where effectiveness is judged by whether some measure of short-term (hourly-weekly) variability in the data is reduced through removal of identified outliers. Our short-term variability measure is determined by fitting a smooth curve to the hourly data (Section 5.3), subtracting this from the hourly data to give a timeseries of residuals, and then calculating the standard deviation of the residuals. Figure 8 shows that as the $CO_2$ SD rejection value is reduced, the residual SD initially decreases rapidly

(Fig. 8b) while the proportion of hourly data rejected increases relatively slowly (blue curve, Fig. 8a); at 0.3 $\mu$mol mol$^{-1}$ only about 7% of hours are rejected but the residual SD has been reduced from 0.46 to 0.28 $\mu$mol mol$^{-1}$. The residual SD continues to decrease until the SD rejection value reaches around 0.15 $\mu$mol mol$^{-1}$. The data loss starts to increase more rapidly when





the SD rejection value is less than about 0.19 $\mu mol\,mol^{-1}$; data loss is greater than 80% for a SD rejection value of 0.10 $\mu mol\,mol^{-1}$.

Figure 8a also shows the proportion of rejected data that are outliers (here taken as the magnitude of a residual being greater than 0.5 $\mu mol\,mol^{-1}$). As the maximum minute SD rejection threshold is reduced the proportion of outliers rejected becomes

smaller; by 0.24 $\mu mol\,mol^{-1}$ the selection is removing as many low residual data points as outliers (red curve, Fig. 8a). The analysis suggests that a SD rejection value between 0.15–0.20 $\mu mol\,mol^{-1}$ provides the best compromise between minimising residual spread and minimising data loss. Figure 9a shows average "hourly" mole fraction from 2006–2017 selected using a maximum minute SD rejection threshold of 0.20 $\mu mol\,mol^{-1}$ compared to all hourly values (selected only for no missing minutes). The selection tends to remove positive outliers throughout the year and some negative outliers in summer. This would

be consistent with the removal being mostly of measurements influenced by local anthropogenic fluxes with a smaller influence from the biosphere on Macquarie Island.

## 5.2   Removing Southern Hemisphere land flux influences

Figure 4c,e demonstrates that elevated radon concentrations are a good indicator of air samples that have been influenced by long-range transport from Southern Hemisphere continents. Radon observations are not available for the whole period of

LoFlo $CO_2$ measurements at Macquarie Island with radon observations only commencing in 2011. The early portion of this data record, 2011 until March 2013, are also of poor quality due to an instrumental issue that could only be addressed at the annual resupply visit. For this reason, to ensure that the $CO_2$ record (2005–2016) is treated consistently, we test the feasibility of using model-simulated radon concentrations instead of the observed radon concentrations. Where the records overlap, the modelled radon is broadly consistent with the observations but with generally lower baseline concentrations. This means that

the analysis presented here, to choose an appropriate radon selection threshold, is applicable to this modelled radon data set only and would need to be repeated if using the available observations or an alternative modelled radon data set.

Atmospheric radon concentrations are simulated as in Loh et al. (2015), except that the CSIRO Conformal-Cubic Atmospheric Model (McGregor, 2005; McGregor and Dix, 2008) is nudged to ECMWF winds (Dee et al., 2011) rather than the NCEP forcing used previously. Radon is input to the lowest model level at a constant rate of $1.66\times10^{-20}$ $mol\,m^{-2}\,s^{-1}$ (21.0

$mBq\,m^{-2}\,s^{-1}$) for land surfaces and $8.30\times10^{-23}$ $mol\,m^{-2}\,s^{-1}$ (0.11 $mBq\,m^{-2}\,s^{-1}$) for ocean surfaces between 60°S and 60°N, of $8.30\times10^{-23}$ $mol\,m^{-2}\,s^{-1}$ for both land and ocean between 60 and 70°N and S, and zero poleward of 70°. Following injection, radon decays with a half-life of 3.8 days. Here we report hourly radon concentrations output from the model at the nearest grid-cell to Macquarie Island (159.229°E, 54.854°S).

The effectiveness of selection by radon threshold is assessed in Fig. 8c,d, starting from the case shown in Fig. 9a where local

impacts have been removed using the maximum minute $CO_2$ SD criteria of 0.2 $\mu mol\,mol^{-1}$. Radon selection clearly reduces the residual standard deviation (Fig. 8d) below the minimum standard deviation from local selection alone (0.26 $\mu mol\,mol^{-1}$) to 0.20 $\mu mol\,mol^{-1}$ for a radon threshold of 60–90 $mBq\,SCM^{-1}$ and to 0.16 $\mu mol\,mol^{-1}$ for radon of 20 $mBq\,SCM^{-1}$. As with the local selection there is a compromise between reducing residual standard deviation and maintaining data quantity. The proportion of rejected data reaches 0.40 for a radon rejection value of 60–70 $mBq\,SCM^{-1}$ and increases rapidly as the radon




rejection value is further reduced (Fig. 8c). Using the same outlier measure as in the previous section, only around a third or less of the additional data points rejected by radon selection, relative to local selection, are outliers. This may be because the model-simulated radon is likely to give a more diffuse signal than observations and hence would reject more data. It is also possible that air with a radon signal has traversed a continental region with little or no $CO_2$ flux (e.g. little vegetation cover or

at a time of day/year when fluxes are small) and consequently is not seen as an outlier at Macquarie Island.

Fig. 9b shows the impact of radon selection at 60 mBq SCM$^{-1}$ relative to maximum minute $CO_2$ SD selection at 0.2 µmol mol$^{-1}$. Both positive and negative outliers are removed, with negative outliers more prominent in spring.

### 5.3   Curve fitting

A smooth curve was fit to the hourly $CO_2$ data following the methods described in Thoning et al. (1989). The first step is to

fit the data (using least squares) with a 2nd degree polynomial and four harmonics to represent the long-term increase in $CO_2$ and a mean seasonal cycle. While many applications of the Thoning et al. method iterate this fit, removing outliers after each iteration, this was not required for the Macquarie Island dataset. Residuals from the polymonial+harmonic fit were then filtered in the frequency domain, with transformation to the frequency domain using a sampling interval of 1 hour. Two filters were applied to capture short and long-term variations. An 80-day low-pass filter (which retains variability on weekly to monthly

timescales, providing a filter which captures interannual variations in seasonality), and a 667 day low-pass filter (which captures interannual variations in $CO_2$ growth that are not represented by the 2nd degree polynomial). Either set of filtered residuals are combined with all or part of the polynomial+harmonic fit to represent different features of the $CO_2$ timeseries.

Figure 9c shows the smooth curve fit to the hourly Macquarie Island $CO_2$ observations from combining the polynomial+harmonic with the 80-day filtered residuals, for the three datasets shown in Fig. 9a,b. The three cases are difficult to

distinguish, confirming the relatively small number of outliers observed at Macquarie Island and their small influence on the fitted curve. Figure 9c also shows the difference in the fitted curves (right-hand axis) from the fit to the dataset selected for both minute $CO_2$ SD and radon. Differences are mostly positive and up to 0.18 µmol mol$^{-1}$ for the fit to the dataset selected only for no missing minutes, consistent with this dataset having mostly positive outliers. Differences are smaller and more centred on zero for the fit to the dataset with minute $CO_2$ SD selection. Although the differences between the curve fits are small, data

selection remains important because $CO_2$ gradients across the Southern Ocean are also small.

### 6   Macquarie Island baseline $CO_2$ climatology

Using the maximum minute $CO_2$ SD (0.2 µmol mol$^{-1}$) and radon (60 mBq SCM$^{-1}$) selected dataset as the baseline LoFlo record for Macquarie Island, we briefly present the main features of the baseline climatology compared to that derived previously from flask measurements.



## 6.1   Long-term trend and growth rate

The long-term trend in Macquarie Island LoFlo $CO_2$ is represented by the sum of the 2nd degree polynomial fit and the 667-day filtered residuals. This is shown in Fig. 10a along with an equivalent fit to the Macquarie Island flask measurements. The long-term trends are very similar with a gradual increase in baseline $CO_2$ concentrations over the 8 year period from 377 to 392

$\mu mol\, mol^{-1}$ . The derivative of the long-term trend, the $CO_2$ growth rate, is shown in Fig. 10b and here the subtle differences in the long-term trend between the LoFlo and flask records become more evident. From 2006–2010 the growth rate from the flask record is less variable than from the LoFlo record, while there is much better agreement for the 2010–2013 period. Figure 7 showed that around 2008 the differences between flask and LoFlo measurements were more negative than for other periods, coincident with generally longer storage times and hence larger storage corrections. It is possible that a small bias in flask

measurements through 2008 is sufficient to influence the derivative of the long-term trend through 2007–2009. This highlights the sensitivity of the growth-rate calculation to small, systematic biases in observed mole fraction, to which the flask record is much more susceptible than the in-situ LoFlo record.

## 6.2   Seasonal cycle

The seasonal variation in $CO_2$ at Macquarie Island (Fig. 10c) is conventionally revealed by removing the long-term trend

curve from the curve fit that combines the fitted polynomial+harmonics with the 80-day filtered residuals. The seasonal cycle has a peak to trough amplitude of around 1.5 $\mu mol\, mol^{-1}$ with a minimum around February-March and a maximum around October. There are interannual variations in the seasonality, perhaps more in amplitude than phase, that are mostly picked up in both the LoFlo and flask records. The LoFlo produces a much smoother representation of the seasonal cycle than the flask record. This is due both to the higher precision of the data and its much higher temporal frequency. The comparison clearly

shows the limitations of the MQA flask data. Despite the very clean Southern Ocean environment, the combination of small but unresolved synoptic variability in $CO_2$ mole fraction and the necessity of making large storage corrections to the flask data, mean that even smooth curve fits to the quality-controlled flask record contain unrealistic features that could easily be misinterpreted. The LoFlo record provides a much more reliable representation of the seasonality of atmospheric $CO_2$ over the Southern Ocean.

It is important to note that interpretation of the interannual variations in the MQA LoFlo seasonality cannot only consider interannual variations in Southern Ocean fluxes. Tropical and Northern Hemisphere fluxes also make a significant contribution to seasonality across the Southern Ocean (e.g. Law et al., 2006) the magnitude and timing of which will be influenced by interannual variability in interhemispheric transport (e.g. Francey and Frederiksen, 2016). While this remote contribution complicates the interpretation of the seasonality at a single Southern Ocean site, such as Macquarie Island, comparisons between

the seasonality of different Southern Ocean sites may be more revealing. The high precision of the MQA LoFlo record, the equivalent LoFlo record at Cape Grim and the cavity ring-down spectroscopic records at Casey Station, Antarctica now make these across-Southern Ocean comparisons possible and this is a focus of ongoing research.





## 7 Conclusions

The Southern Ocean plays a key role in the global $CO_2$ cycle but studies investigating the variability and seasonality of the sink have been limited by the paucity of both atmospheric and oceanic $CO_2$ data in the region with sufficient precision to resolve the small but large scale atmospheric variation. The observations presented here are a new data stream from a key

location within the Southern Ocean region that can contribute to the investigation of Southern Ocean $CO_2$ flux variability and atmospheric transport. Estimates of the uncertainty associated with this record are typically small and dependent on the intended end application of the data set. They vary with the temporal averaging period, the network choice and the magniude of the sample-reference difference. For applications that compare LoFlo datasets within the CSIRO network, the uncertainty on 30-minute mean samples with mole fractions near the reference standard (> 99.9 % of all observations) is 0.025 $\mu$mol mol$^{-1}$,

allowing reliable measurement of spatial gradients across the Southern Ocean.

The in-situ nature of this record (unlike the traditional flask measurements) results in an increase in the temporal frequency of the data and hence a far richer data stream. The in-situ record and its statistically derived products (baseline, growth rate, long-term trend and seasonality) are more robust than those of the co-located flask record as the impediments of long sample storage times are removed and the temporal frequency of the observations is increased by multiple orders of magnitude. The

increased temporal frequency has revealed diurnal and synoptic variations in atmospheric $CO_2$ at Macquarie Island which will be explored further in future work. In particular, the combination of this record with other high-precision in-situ sites will allow the quantification of small but significant spatial gradients across the Southern Ocean.

## 8 Code availability

The fortran version of the curve fitting code used in this paper is not publically available, however, a C language program

version can be found at ftp://ftp.cmdl.noaa.gov/user/thoning/ccgcrv/. CCAM is an open source model. Information about the model and installation can be found at https://confluence.csiro.au/display/CCAM/CCAM and the code accessed directly at https://bitbucket.csiro.au/projects/CCAM.

## 9 Data availability

The MQA LoFlo $CO_2$ data set is currently being prepared for submission to the World Data Centre for Greenhouse Gases.

Radon data is available at https://www.researchgate.net/publication/327427854_Macquarie_Island_Hourly_Radon_Observations_-2013-2016.

*Author contributions.* ARS, RML analysed the data, performed the model simulations and wrote the paper with input from co-authors, ARS, MvS serviced and managed the instrument, MvS installed the instrument, RF led the development of the LoFlo instrumentation and calibration strategy, RLL contributed to the uncertainty analysis and provided the MQA flask data and its analysis, ARS, MvS, DAS, PBK,

RTH contributed to calibration, database and analysis software development, SDC, AGW, SW contributed the radon data.



*Competing interests.* The authors declare that they have no conflict of interest

*Acknowledgements.* The authors would like to thank L. Paul Steele for his simulating discussions and suggestions in relation to this paper and acknowledge his significant involvement in the development and calibration of the LoFlo system and the GASLAB flask sampling program. This research was funded in part by the Australian Government Department of the Environment, the Bureau of Meteorology, and CSIRO

5   through the Australian Climate Change Science Programme and directly by CSIRO. The authors would also like to acknowledge the in-kind support of the Australian Antarctic Division, under project no. 4167 – Greenhouse gases in the southern atmosphere, and the Australian Bureau of Meteorology. CCAM modelling was undertaken on the NCI National Facility in Canberra, Australia, which is supported by the Australian Commonwealth Government. Back-trajectories were calculated using the HYSPLIT transport and dispersion model from NOAA Air Resources Laboratory. Ot Sisoutham has provided support to the Macquarie Island and Southern Ocean radon program. Maps used in

10   Fig. 1 are courtesy of the Australian Antarctic Division.





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



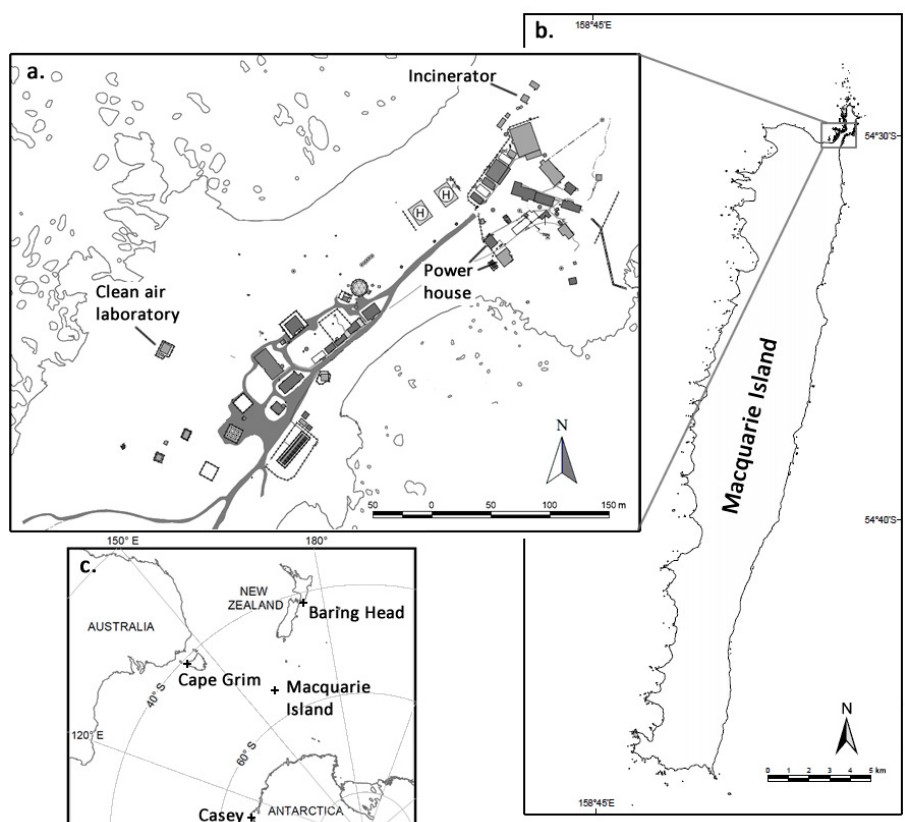

**Figure 1.** Macquarie Island isthmus map (a) showing the position of the clean air laboratory, power houses, incinerator and other station buildings; Macquarie Island whole island map (b) and Southern Ocean in-situ $CO_2$ measurement stations (c).




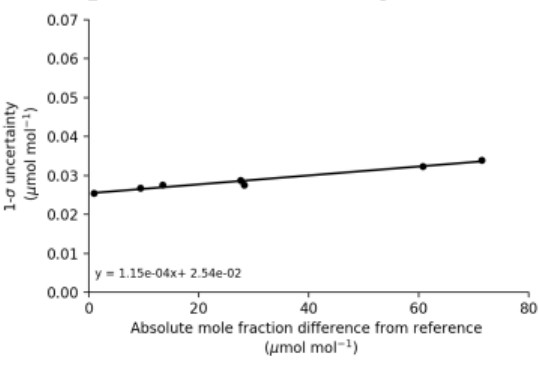
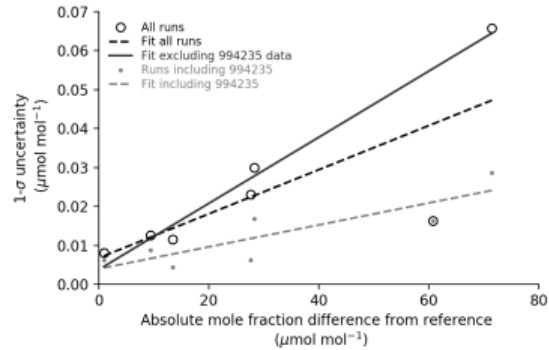

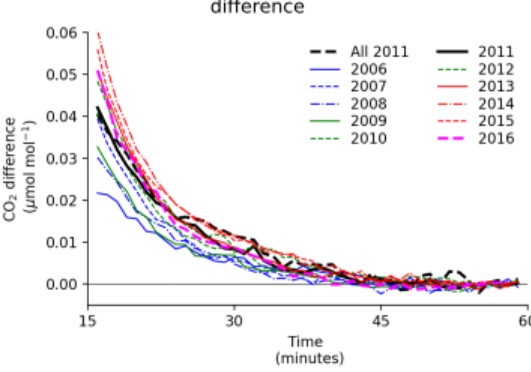
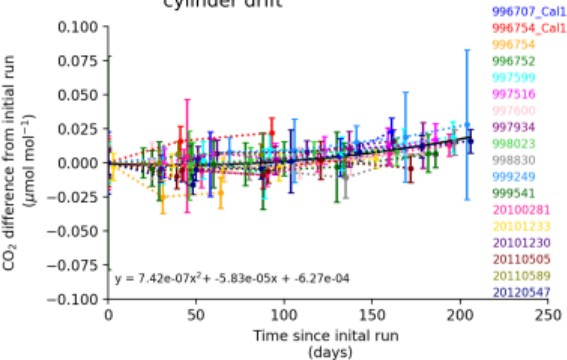

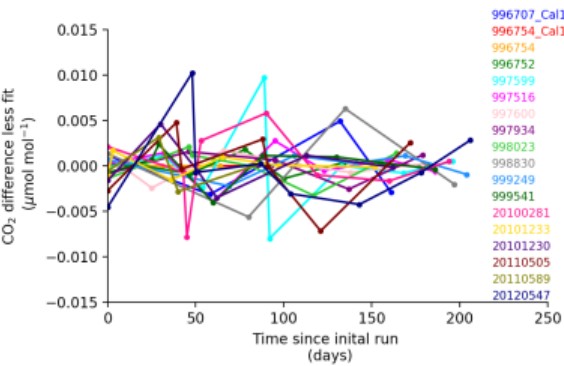
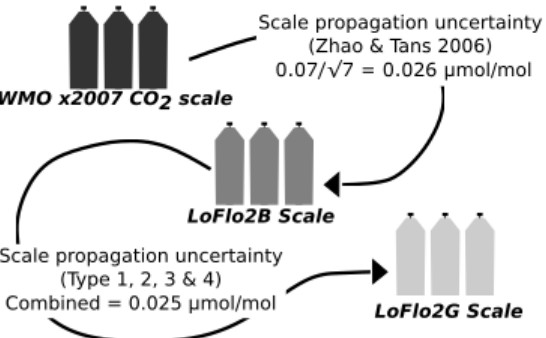

**Figure 2.** Error components of the Macquarie Island data site.



Figure 2 caption (cont.)

(a) Mean of the mean standard deviation of the 1 minute mole fractions calculated for each run of each calibration cylinder as determined using the nonlinearity correction of the previous run (filled circles) and a linear fit to these data (solid line). (b) Mean standard deviation of the difference between calibration cylinder mole fractions determined for each individual run and the mean cylinder mole fraction of all

5   runs (open circles) and of runs that included cylinder 994235 (closed circles). A linear fit to all runs (dashed black line), a linear fit to runs that included cylinder 994235 (dashed grey line) and a linear fit to the all runs data excluding the 994235 data point (solid black line) are also shown. (c) Mean minute $CO_2$ mole fraction difference from the mixing ratio averaged over minutes 55–59, for 6337 available hours in 2011 with 44 minutes of sampling (black, dashed) and for hours with $CO_2$ standard deviation less than $0.15\ \mu\mathrm{mol\ mol}^{-1}$ for all 44 minutes in the hour for each year as listed in the key. (d) Long term drift in reference cylinder mole fraction over time for each reference cylinder

10  as referenced in the key. (e) Short term variability in reference cylinder mole fraction over time determined as the difference between the individual drift values of each cylinder and a quadratic fit to these values for each reference cylinder as listed in the key. (f) Scale propagation chain and an estimate of the associated scale propagation error for each step.





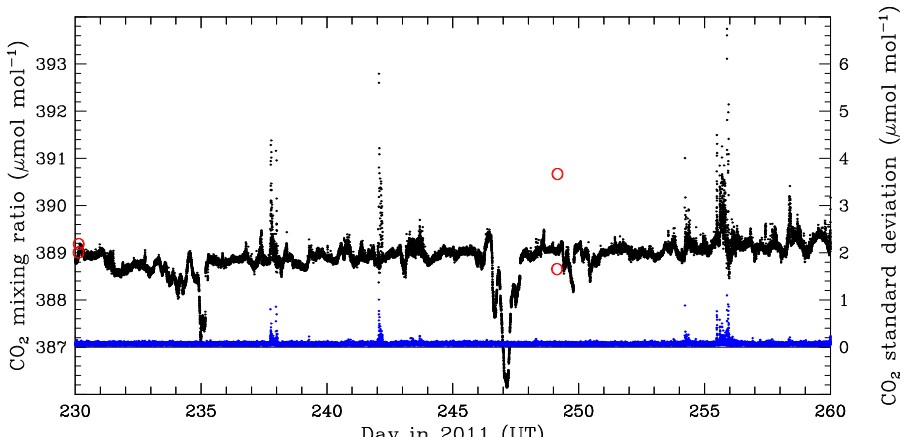

**Figure 3.** Minute mean (black dot, left axis) and standard deviation (blue dot, right axis) of $CO_2$ mole fraction for days 230–260 (August 18 to September 17) in 2011. Red open circles at day 230 and day 249 are $CO_2$ mole fraction from flask samples.





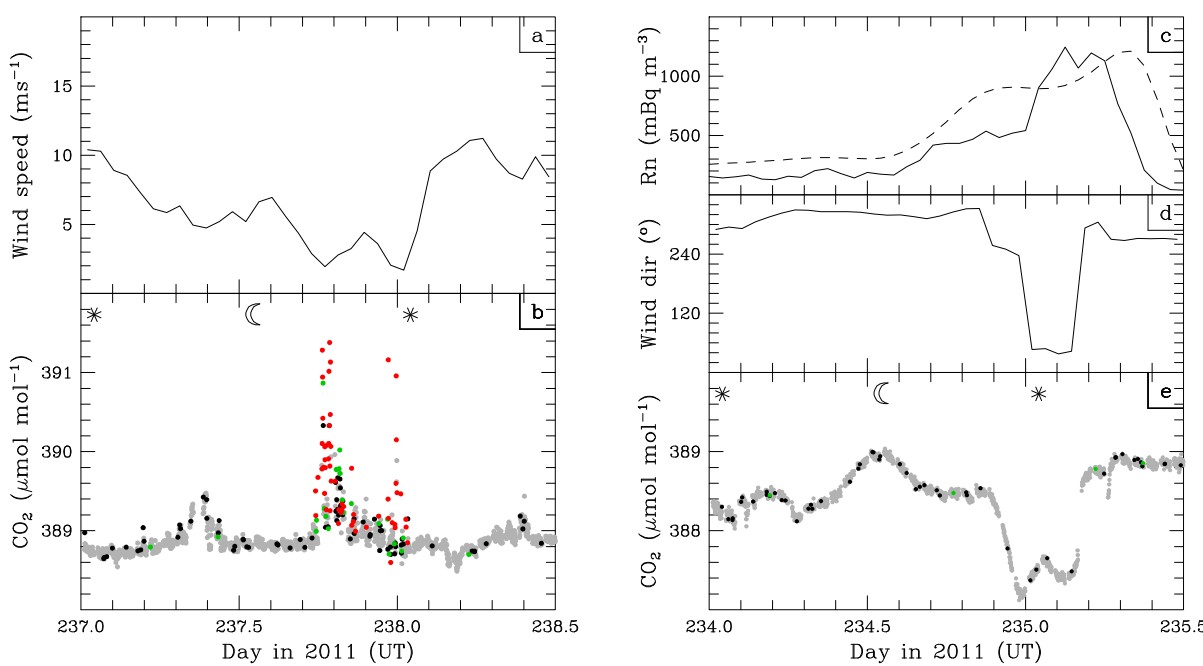

**Figure 4.** Hourly wind speed (a) and minute mean $CO_2$ mixing ratio (b) for days 237–238.5 (Aug 25 0:00 UT to Aug 26 12:00 UT) in 2011. Hourly observed (solid) and modelled (dashed) radon concentration in $\mathrm{mBq\,m^{-3}}$ (c), wind direction (d) and minute mean $CO_2$ mole fraction (e) for days 234–235.5 (Aug 22 0:00 UT to Aug 23 12:00 UT) in 2011. $CO_2$ mixing ratio is coloured according to $CO_2$ standard deviation: less than $0.10\,\mathrm{\mu mol\,mol^{-1}}$ (grey), $0.10$–$0.12\,\mathrm{\mu mol\,mol^{-1}}$ (black), $0.12$–$0.15\,\mathrm{\mu mol\,mol^{-1}}$ (green), greater than $0.15\,\mathrm{\mu mol\,mol^{-1}}$ (red).





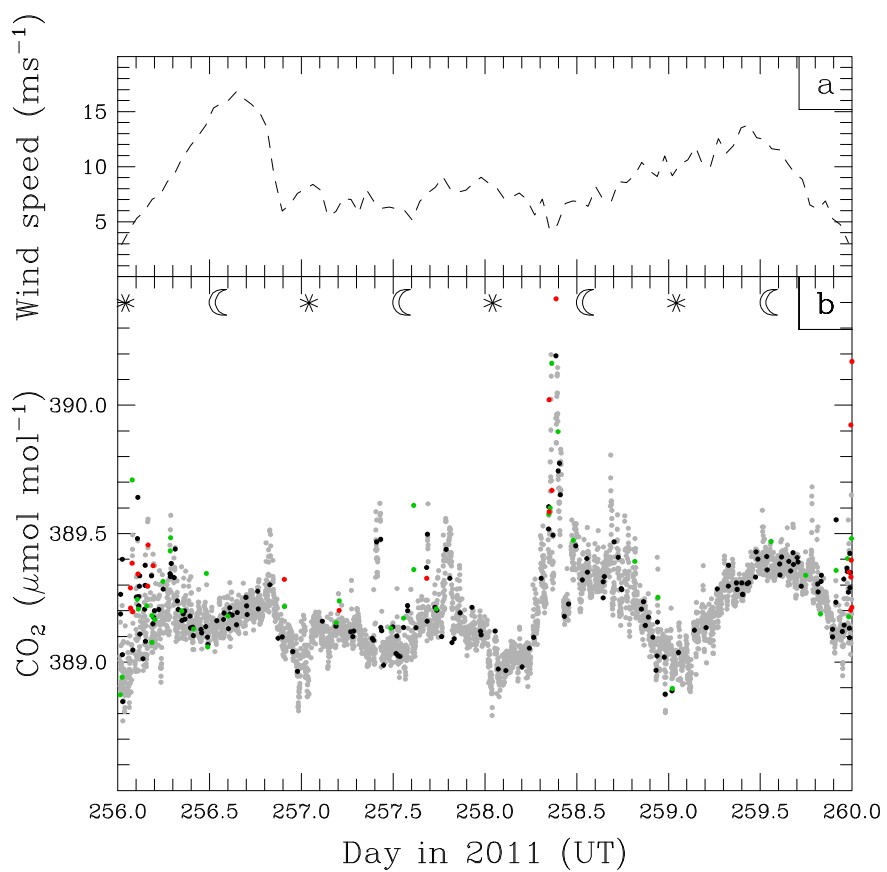

**Figure 5.** Hourly wind speed (a) and minute mean $CO_2$ mixing ratio (b) for days 256–260 (Sep 13 0:00 UT to Sep 17 0:00 UT) in 2011. $CO_2$ mole fraction is coloured according to $CO_2$ standard deviation: less than $0.10\,\mathrm{\mu mol\,mol^{-1}}$ (grey), $0.10$–$0.12\,\mathrm{\mu mol\,mol^{-1}}$ (black), $0.12$–$0.15\,\mathrm{\mu mol\,mol^{-1}}$ (green), greater than $0.15\,\mathrm{\mu mol\,mol^{-1}}$ (red).





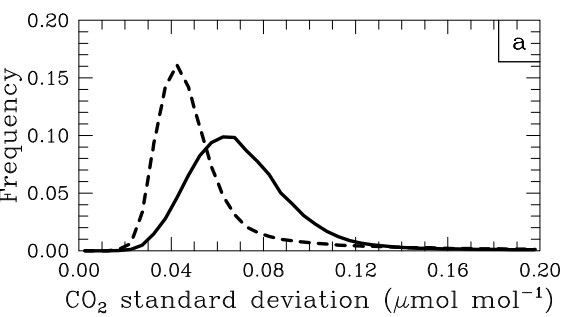
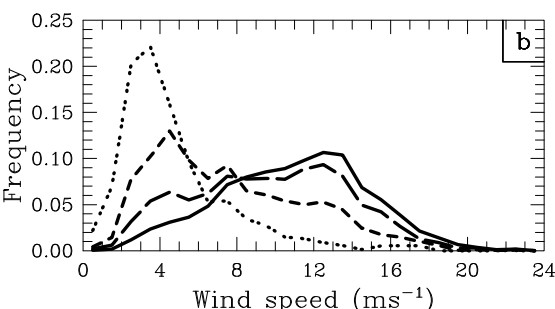

**Figure 6.** Frequency histograms of (a) the standard deviation of $CO_2$ mole fraction within a minute for all available minutes in 2011 for Macquarie Island (solid) and Cape Grim (dashed) and (b) the nearest hourly wind speed to all available minutes in 2011 with $CO_2$ standard deviation less than $0.10\,\mu\mathrm{mol\,mol}^{-1}$ (solid), $0.10$–$0.12\,\mu\mathrm{mol\,mol}^{-1}$ (long dash), $0.12$–$0.15\,\mu\mathrm{mol\,mol}^{-1}$ (short dash) and greater than $0.15$ $\mu\mathrm{mol\,mol}^{-1}$ (dotted).





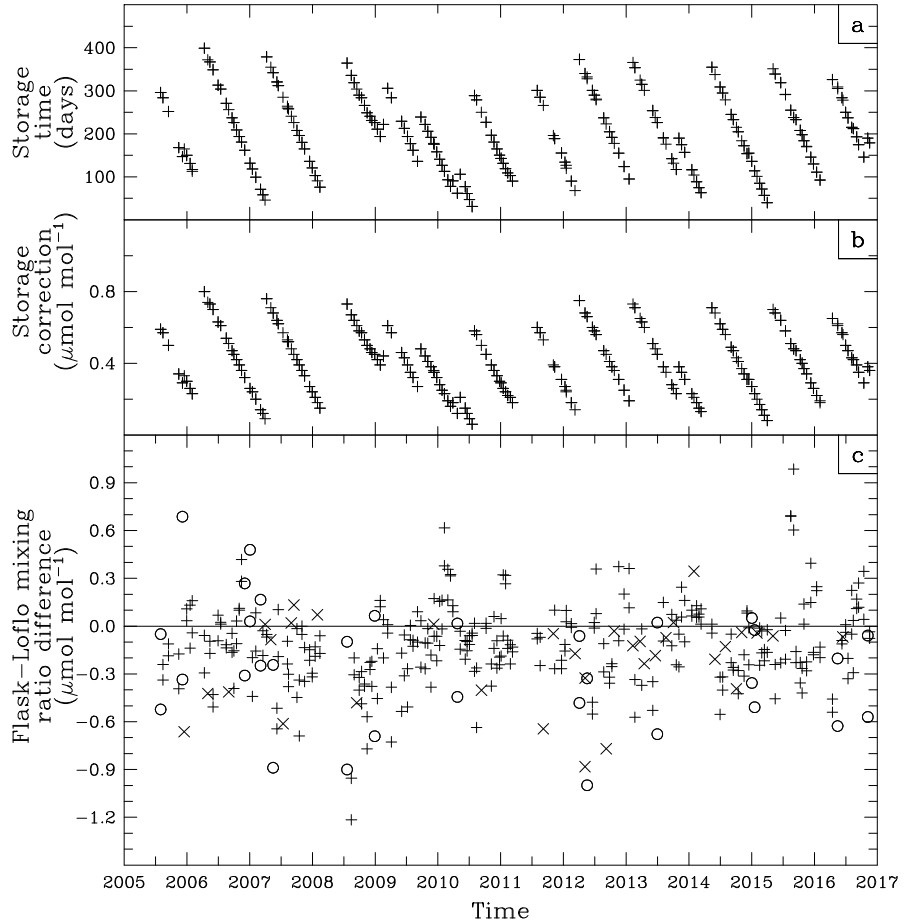

**Figure 7.** Storage time (a) in days, storage correction (b) in $\mu\mathrm{mol\,mol^{-1}}$, for flasks filled at Macquarie and the mole fraction difference (c) between the flask mole fraction and the mean Loflo2 mole fraction for the two 30-minute averages that span the flask (or if both not available, the single 30-minute average within 1 hour of the flask fill time). Mole fraction differences are shown for all flasks that had a flask pair difference less than $0.4\ \mu\mathrm{mol\,mol^{-1}}$ (+), for all flasks that had a flask pair difference greater than $0.4\ \mu\mathrm{mol\,mol^{-1}}$ (o) and for flasks without a pair (x). Flagged flasks are not used.





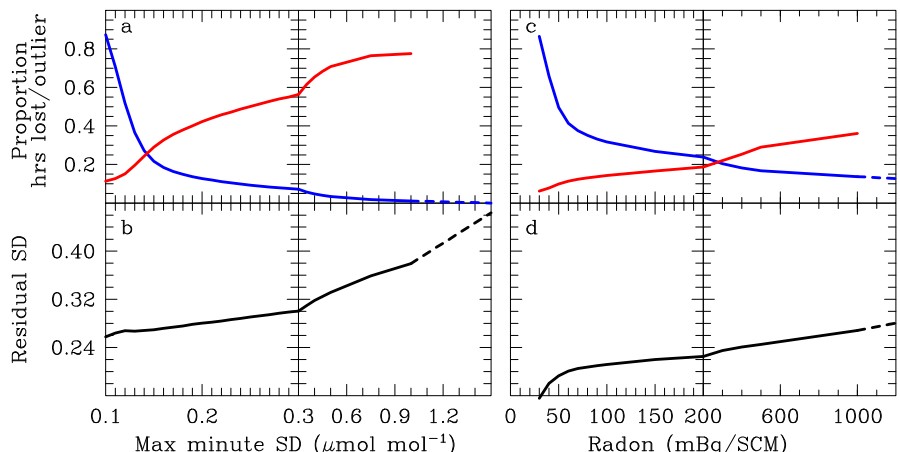

**Figure 8.** Proportion of hours lost (blue) and proportion of rejected hours that are outliers (red) for a given minute $CO_2$ standard deviation rejection criteria (a) and for a given radon rejection criteria in addition to maximum minute $CO_2$ SD selection of $0.2\,\mu\text{mol}\,\text{mol}^{-1}$ (c). Outliers are defined as hours with mole fraction residual from a smooth curve fit greater than $0.5\,\mu\text{mol}\,\text{mol}^{-1}$. Standard deviation of residuals from a smooth curve fitted to data selected by the maximum minute $CO_2$ SD (b) and selected by radon concentration in addition to maximum minute $CO_2$ SD of $0.2\,\mu\text{mol}\,\text{mol}^{-1}$ (d).





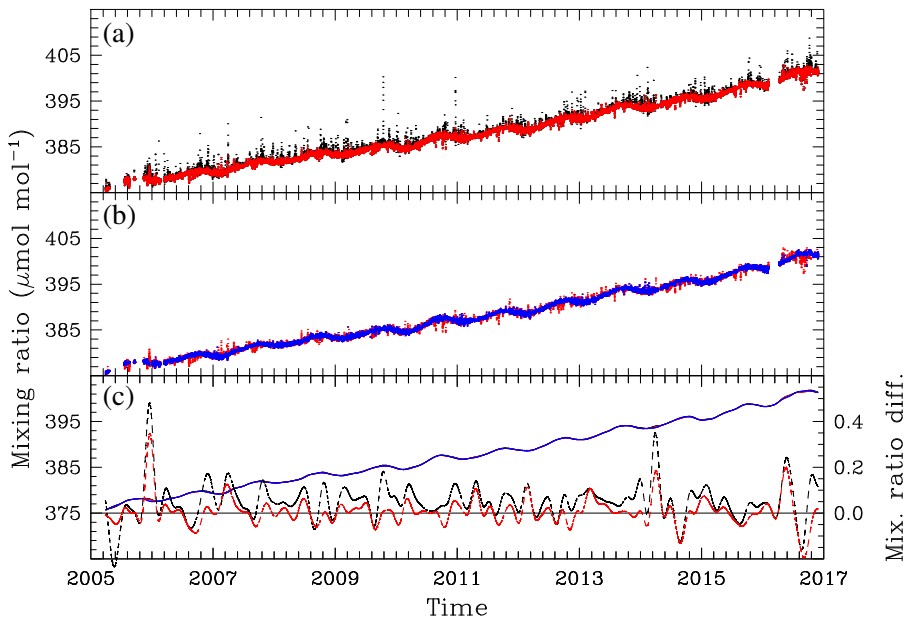

**Figure 9.** $CO_2$ mole fraction ($\mu mol\,mol^{-1}$) at hourly frequency (a,b) and fitted with a smooth curve (c). Hourly frequency data are 30-minute means where the 30-minute means have been selected only for no missing minutes (black, a), additionally for maximum minute $CO_2$ standard deviation (SD) less than 0.2 $\mu mol\,mol^{-1}$ (red,a,b) and additionally for model simulated radon concentration less than 60 $mBq\,SCM^{-1}$ (blue,b). Panel (c) shows the curve fits for each of the three plotted datasets (solid, colours as panel a,b) and the difference in the curve fit (right axis) from the radon-selected fit for missing data only selection (dotted, black) and for maximum minute $CO_2$ SD selection (dotted, red).





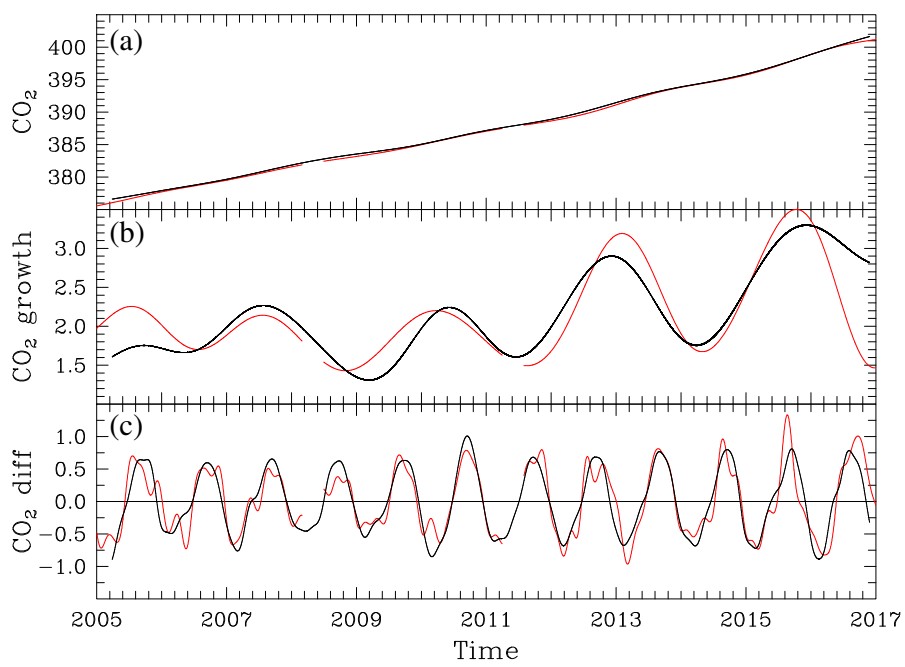

**Figure 10.** Macquarie Island $CO_2$ long-term trend in $\mu mol\,mol^{-1}$ (a), growth rate in $\mu mol\,mol^{-1}y^{-1}$ (b) and with the trend removed (c) using fits to hourly frequency LoFlo data (black) and flask data (red).



**Table 1.** Calibration cylinder concentrations on WMO X2007 scale, as measured by LoFlo2B or GASLAB. Suite 2G-a was used from 2005 to April 2006, Suite 2G-b from April 2006 to the present.

| Cylinder No. | Suite 2G-a | | Suite 2G-b | |
| --- | --- | --- | --- | --- |
| | LoFlo2B | GASLAB | LoFlo2B | GASLAB |
| 1 | 355.00±0.02 | 355.1±0.1 | 317.64±0.02[1] | 317.65±0.06[1] |
| 2 | 363.19±0.01 | 363.24±0.07 | 356.46±0.01 | 356.55±0.06 |
| 3 | 372.16±0.02 | 372.23±0.04 | 370.63±0.01 | 370.74±0.06 |
| 4 | 383.45±0.02 | 383.49±0.04 | 385.01±0.01 | 385.09±0.08 |
| 5 | 399.72±0.03 | 399.75±0.06 | 393.49±0.01 | 393.59±0.06 |
| 6 | 415.22±0.03 | 415.2±0.1 | 412.25±0.01 | 412.30±0.08 |
| 7 | 429.29±0.04 | 429.20±0.05 | 455.45±0.01 | 455.57±0.15 |

1: Only used April 2006 to March 2009



**Table 2.** Combined uncertainty estimates in $\mu mol\,mol^{-1}$ applicable to comparisons with different datasets for minute mean data and hourly data based on averaging the final 30 minutes of each hour. The uncertainty estimates are given as a range spanning sample to reference differences of 0 - 10 $\mu mol\,mol^{-1}$

| Averaging period | LoFlo2G internal | CSIRO high-precision network | WMO X2007 networks |
|---|---|---|---|
| Minute | 0.033 - 0.034 | 0.045 - 0.046 | 0.052 - 0.053 |
| 30 minute | 0.007 - 0.010 | 0.025 - 0.027 | 0.036 - 0.037 |