# Peer review of "The Macquarie Island [LoFlo2G] high-precision continuous atmospheric carbon dioxide record"

_Atmospheric Measurement Techniques, 2018_

## Referee Comment (RC1)

**Review of Stavert et al: The Macquarie Island [LoFlo2G] high-precision continuous atmospheric carbon dioxide record for AMT**

**November 22, 2018**

The article by Stavert et al is submitted to Atmospheric Measurement Techniques (AMT). It describes retrieved CO2 measurements with a LoFlo2 instrument from Macquarie Island, a site in the Southern Ocean and the importance of this unique dataset.

The paper gives a detailed description about the site at Macquarie Island, measurement collection routines and limitations, the instrument setup, calibration and uncertainty analysis, definition of the baseline record as well as a general climatology of the dataset.

Main comments:

- 1. It's probably too late now, but I would suggest that this is not so much a "technique" paper as a "data" paper, and ESSD(D) might have been a better target journal.
- 2. I would highlight the importance of these measurements even more in the introduction, and the potential 'gaps' these measurements could fill with references about studies that focused on the importance of Southern Ocean CO2 measurements and their application.
- 3. If it is not too time consuming a paragraph about general error prorogation (with reference) and adding the difference between those and your measurement uncertainty method would be useful. More emphasis on the filtering techniques would lean the paper back towards AMT appropriate.
- 4. A few sections in the results could be simplified (e.g. the discussion of using minutely S.D. to filter out local influences). A careful reading to condense some of the text would be useful.
- 5. In terms of the uncertainties (e.g. Type 4) have you tested using the interquartile range (or the 25th and 75th percentile) as the measure of uncertainty instead of the 1 sigma, and also maybe to weight the fitting based on the uncertainty? This is more a comment and I am not suggesting to re-calculate everything but it would be interesting (maybe in some future measurement uncertainty quantification work) to see how much those changes would affect the results.

In general, the paper is nicely written, scientifically sound and worthy of publication. After addressing these and other minor comments the manuscript will be suitable for publication.

**1 General Comments**

- A number of abbreviations are not defined the first time they are mentioned (e.g. MQA in abstract and thoughout the text, CSIRO in the introduction, WMO). Also you jump from writing the full term to abbreviations often, it would be better to have some consistency, either use the full term of the abbreviation.
- Page 2 line 12 However, efforts...  $\rightarrow$  is there some additional reference for this sentence/statement?
- Page 2 line 16 subantartic zone and polar front zone  $\rightarrow$  are there any studies that explore how this affects the measurements?

**2 Technical Comments**

- Page 2 line 7  $\rightarrow$  The Southern Ocean abbreviation (SO) is unnecessary, it is only used in the introduction.
- Page 3 line 5-6 north-south and south east  $\rightarrow$  consistency, do you need a dash or not?
- Page 13 line 1 'the figure'  $\rightarrow$  specify again which figure
- Page 14 line 15 criterion .  $\rightarrow$  remove the space before the dot
- Page 14 line 16 Standard deviation (SD) → you used the standard deviation before in the text so define the abbreviation before.
- Page 16 line 11 Thoning et al.  $\rightarrow$  missing year
- Figure 3, could the right axis (standard deviation) be coloured to blue? Do the flask samples come with some uncertainty that could be added to the plot?

---

## Referee Comment (RC2)

**Review report**

Journal : AMT

MS No. : amt-2018-300

Title : **The Macquarie Island [LoFlo2G] high-precision continuous atmospheric carbon dioxide record**

Author(s) : Ann R. Stavert et al.

The authors report on high-precision continuous measurements of carbon dioxide (CO2) in the framework of the Commonwealth Scientific and Industrial Research Organisation (CSIRO). The measurements were performed at Macquarie Island in the Southern Ocean region using a LoFlo2 instrument based on non dispersive infrared (NDIR) method. A detailed discussion is given on protocols of measurements and calibration, analysis of measurement uncertainty sources and procedures for baseline determination.

**General comments**

The manuscript is well structured and written, scientifically sound. It would be acceptable for publication in AMT after minor revision by addressing the comments and questions listed below.

**Specific comments**

(1) Page 14, section 5 "Defining a baseline record" : the authors discussed how to remove influences of local flux and Southern Hemisphere land flux to achieve a regional background CO2 observation. Should be considered the ocean-atmosphere CO2 exchange in this case? If yes, how to take into account such influence?

(2) Page 16, section 5.3 "Curve fitting" : a low-pass filter is used to fit the hourly CO2 data to smooth the time-series results. Is the Kalman filtering method more suitable for such application?  As an adaptive filtering technique, Kalman filter can efficiently remove the shot-to-shot variability related to the real-time noise in the measured data with minimal deformation of the physical quantity to be measured. Kalman filtering method has been successfully applied to perform fast and high-precision measurements of trace gas concentration [*Appl. Phys. B* **74** (2002) 85-93] and isotope ratio [*Opt. Lett.* **35** (2010) 634–636].

**Technical corrections**

(1) Page 6, line 8 and line 28 : "Fig. 2e" should be "Fig. 2f"?

(2) Please make larger the following figures 2, 4, 6 and 7.

(3) Some sentences are too long. It would be better to rephrase them and to make them more understandable. For instance,

Page 2, lines 27-30 : "While the performance .... to reduced calibration requirements";

Page 5, lines 24-26 : "Macquarie Island .... sample measurements";

Page 8, lines 24-26 : "This is independent .... in a calibration run)";

Page 10, lines 2-4 : "The uncertainty .... 18 reference standards";

Page 15, lines 21-23 : "Radon is input .... and zero poleward of 70°";

Page 15, lines 27-29 : "Radon selection .... for radon of 20 mBqSCM-1";

Page 17, lines 21-23 : "Tropical and northern .... Frederiksen, 2016)";

Page 18, lines 7-9 : "The in-situ .... by multiple orders of magnitude";

....

---

## Author Comment (AC1) · 15 Jan 2019

**Authors Response Reviewer 1 - The Macquarie Island [LoFlo2G] high-precision continuous atmospheric carbon dioxide record**

Ann R. Stavert[1], Rachel M. Law[1], Marcel van der Schoot[1], Ray L. Langenfelds[1], Darren A. Spencer[1], Paul B. Krummel[1], Scott D. Chambers[2], Alistair G. Williams[2], Sylvester Werczynski[2], Roger J. Francey[1], and Russell T. Howden[1]

[1] CSIRO Oceans and Atmosphere, Aspendale, Victoria, 3195, Australia

[2] Australian Nuclear Science and Technology Organisation, Kirawee, New South Wales, 2232, Australia

*Correspondence to*: Ann R. Stavert (ann.stavert@csiro.au)

We would like to thank the reviewer for their comments. Below we have provided a response to each comment along with any subsequent changes to the manuscript. The original review is in italics and our responses are in normal font.

*The article by Stavert et al is submitted to Atmospheric Measurement Techniques (AMT). It describes retrieved $CO_2$ measurements with a LoFlo2 instrument from Macquarie Island, a site in the Southern Ocean and the importance of this unique dataset.*
*The paper gives a detailed description about the site at Macquarie Island, measurement collection routines and limitations, the instrument setup, calibration and uncertainty analysis, definition of the baseline record as well as a general climatology of the dataset.*
*Main comments:*

1. *It's probably too late now, but I would suggest that this is not so much a "technique" paper as a "data" paper, and ESSD(D) might have been a better target journal.*

As noted by the reviewer this paper does present a large volume of information in relation to the Macquarie Island dataset and as such, could be considered a "data paper". However, we feel that the description of the LoFlo instrument along with the novel baseline assessment method and the method development work conducted to define the data uncertainty estimate are significant "technique" developments and of interest to the AMT readership. Although previous reports and manuals related to the LoFlo have been published, this is the only description available in peer-reviewed literature.

2. *I would highlight the importance of these measurements even more in the introduction, and the potential 'gaps' these measurements could fill with references about studies that focused on the importance of Southern Ocean CO2 measurements and their application.*

The introduction has been changed to include the below.

"However, better quatification is limited by the temporal and spatial availability of observations (both ocean $CO_2$ and atmospheric $CO_2$) across the Southern Ocean region.

For ocean pCO2, techniques exist to extrapolate and map temporally and spatially sparse measurements but these approaches are limited. Recent work (Ritter et al., 2017) found that while often agreeing on the sign of broad scale decadal trends these methods fail to agree on the magnitude, mean values, interannual variability and regional distribution. Atmospheric CO2 measurements can be used to estimate ocean fluxes through an inversion methodology, with the potential advantage that they sample the impact of fluxes over a wider region than would be achieved with oceanic pCO2 measurements. However, most atmospheric measurements from this region are flask samples and previous work (Law et al., 2008) has shown that the Southern Ocean flux trends calculated by inversions are sensitive to atmospheric $CO_2$ data quality. Lenton et al. (2013) also noted that when observational data were sparse, $CO_2$ inversion results were highly sensitive to data quality and the number of regions used in the inversion. As such the addition of a new in situ data record, like that outlined in this paper, should significantly improve future attemps to quantify the Southern Ocean $CO_2$ sink."

3.  *If it is not too time consuming a paragraph about general error propagation (with reference) and adding the difference between those and your measurement uncertainty method would be useful. More emphasis on the filtering techniques would lean the paper back towards AMT appropriate.*

The introduction of the uncertainty section has been altered significantly to encompass this suggestion, see new text below.

"Measurement uncertainty is typically composed of multiple elements and evaluated using a combination of a statistical analysis of replicate measurements (Type A) or based on an alternate source of information (Type B) (Klausen et al., 2016). The individual Type A and Type B components are then combined, usually in quadrature, to determine the overall measurement uncertainty. An example of this model can be found in Andrews et al. (2014) who evaluate in detail the uncertainty associated with tall tower GHG measurements.

It is particularly important to characterise the measurement uncertainty of the MQA record given the small atmospheric signals at mid-high latitudes in the Southern Hemisphere. An earlier study documents the significant impact of measurement errors and biases of LoFlo, conventional NDIR and flask measurements on $CO_2$ growth rate estimation at Cape Grim, another key Southern Hemisphere site (Francey et al., 2010). Here, following the approach discussed earlier, we aim to quantify the measurement uncertainty of the MQA $CO_2$ observations by examining each of five possible sources of error. We will examine how these errors contribute to the uncertainty of hourly and minutely mean values and combine them to determine estimates of the overall measurement uncertainty."

For clarity we have also changed the section heading from "Error propagation" to "Uncertainty analysis".

> 4. *A few sections in the results could be simplified (e.g. the discussion of using minutely S.D. to filter out local influences). A careful reading to condense some of the text would be useful.*

Agreed. Sec 5.1 and 5.2 have been modified to try and simplify the text. Other smaller changes have been made to remove unnecessary detail.

> 5. *In terms of the uncertainties (e.g. Type 4) have you tested using the interquartile range (or the 25th and 75th percentile) as the measure of uncertainty instead of the 1 sigma, and also maybe to weight the fitting based on the uncertainty? This is more a comment and I am not suggesting to re-calculate everything but it would be interesting (maybe in some future measurement uncertainty quantification work) to see how much those changes would affect the results.*

Using the interquartile range rather than the standard deviation as an estimate of the spread and hence the uncertainty in our measurements is indeed an interesting suggestion. However a preliminary investigation did not find a significant difference between uncertainties calculated using the interquartile range and the standard deviation. We have not looked at weighting the fit and thank the reviewer for the suggestion but considering the size of the within hour variability we do not expect that this will have a large effect on the resulting fit.

**General comments**

*A number of abbreviations are not defined the first time they are mentioned (e.g. MQA in abstract and throughout the text, CSIRO in the introduction, WMO). Also you jump from writing the full term to abbreviations often, it would be better to have some consistency, either use the full term of the abbreviation.*

We have altered the manuscript so that abbreviations are defined the first time they are mentioned and are used more consistently. For MQA, we now introduce this later in the manuscript, specifically noting that this is the station site code for Macquarie Island within the Global Atmosphere Watch regional network. For this reason we mostly restrict our usage of MQA to those parts of the text that relate to the $CO_2$ records at Macquarie Island, while using 'Macquarie Island' in full when discussing the island more generally.

*Page 2 line 12 However, efforts... → is there some additional reference for this sentence/statement?*

This sentence has been altered and an additional paragraph, see below, included in the introduction.

"However, better quatification is limited by the temporal and spatial availability of observations (both ocean $CO_2$ and atmospheric $CO_2$) across the Southern Ocean region.

For ocean pCO2, techniques exist to extrapolate and map temporally and spatially sparse measurements but these approaches are limited. Recent work (Ritter et al., 2017) found that while often agreeing on the sign of broad scale decadal trends these methods fail to agree on the magnitude, mean values, interannual variability and regional distribution. Atmospheric CO2 measurements can be used to estimate ocean fluxes through an inversion methodology, with the potential advantage that they sample the impact of fluxes over a wider region than would be achieved with oceanic pCO2 measurements. However, most atmospheric measurements from this region are flask samples and previous work (Law et al., 2008) has shown that the Southern Ocean flux trends calculated by inversions are sensitive to atmospheric $CO_2$ data quality. Lenton et al. (2013) also noted that when observational data were sparse, $CO_2$ inversion results were highly sensitive to data quality and the number of regions used in the inversion. As such the addition of a new in situ data record, like that outlined in this paper, should significantly improve future attemps to quantify the Southern Ocean $CO_2$ sink."

*Page 2 line 16 subantartic zone and polar front zone → are there any studies that explore how this affects the measurements?*

As this is the first time this data record has been released there are no studies which explore the effects of it's location relative to the subantarctic and polar front zone. As far as the authors know there are also no papers examining this relationship for the co-located flask record. However, as the reviewer notes this is an interesting topic which we hope to explore in future work.

**Technical Comments**

*Page 2 line 7 → The Southern Ocean abbreviation (SO) is unnecessary, it is only used in the introduction.*
As suggested, we have removed this abbreviation.

*Page 3 line 5-6 north-south and south east → consistency, do you need a dash or not?*
We have made these consistent (using a dash in each case).

*Page 13 line 1 'the figure' → specify again which figure*
The figure number (6b) has been added

*Page 14 line 15 criterion . → remove the space before the dot*
The space has been removed.

*Page 14 line 16 Standard deviation (SD) → you used the standard devia- tion before in the text so define the abbreviation before.*
The abbreviation has been moved earlier in the text and used from that point on.

*Page 16 line 11 Thoning et al. → missing year*
This has been added

*Figure 3, could the right axis (standard deviation) be coloured to blue? Do the flask samples come with some uncertainty that could be added to the plot?*
The suggested changes have been made and the figure caption updated.

**References**

Andrews, A. E., Kofler, J. D., Trudeau, M. E., Williams, J. C., Neff, D. H., Masarie, K. A., Chao, D. Y., Kitzis, D. R., Novelli, P. C., Zhao, C. L., Dlugokencky, E. J., Lang, P. M., Crotwell, M. J., Fischer, M. L., Parker, M. J., Lee, J. T., Baumann, D. D., Desai, A. R., Stanier, C. O., De Wekker, S. F. J., Wolfe, D. E., Munger, J. W., and Tans, P. P.: $CO_2$, CO, and $CH_4$ measurements from tall towers in the NOAA Earth System Research Laboratory's Global Greenhouse Gas Reference Network: instrumentation, uncertainty analysis, and recommendations for future high-accuracy greenhouse gas monitoring efforts, Atmospheric Measurement Techniques, 7, 647-687, 10.5194/amt-7-647-2014, 2014.

Francey, R. J., Trudinger, C. M., Van Der Schoot, M., Krummel, P. B., Steele, L. P., and Langenfelds, R. L.: Differences between trends in atmospheric CO2 and the reported trends in anthropogenic CO2 emissions, Tellus B, 62, 316-328, doi:10.1111/j.1600-0889.2010.00472.x, 2010.

Klausen, J., Scheel, H. E., and Steinbacher, M.: WMO/GAW Glossary of QA/QC-Related Terminology: https://www.empa.ch/web/s503/gaw_glossary, access: 4/1/2019, 2016.

Law, R. M., Matear, R. J., and Francey, R. J.: Comment on "Saturation of the Southern Ocean CO2 Sink Due to Recent Climate Change", Science, 319, 570a-, 10.1126/science.1149077, 2008.

Lenton, A., Tilbrook, B., Law, R. M., Bakker, D., Doney, S. C., Gruber, N., Ishii, M., Hoppema, M., Lovenduski, N. S., Matear, R. J., McNeil, B. I., Metzl, N., Mikaloff Fletcher, S. E., Monteiro, P. M. S., Rödenbeck, C., Sweeney, C., and Takahashi, T.: Sea–air $CO_2$ fluxes in the Southern Ocean for the period 1990-2009, Biogeosciences, 10, 4037-4054, 10.5194/bg-10-4037-2013, 2013.

Ritter, R., Landschützer, P., Gruber, N., Fay, A. R., Iida, Y., Jones, S., Nakaoka, S., Park, G.-H., Peylin, P., Rödenbeck, C., Rodgers, K. B., Shutler, J. D., and Zeng, J.: Observation-Based Trends of the Southern Ocean Carbon Sink, Geophysical Research Letters, 44, 12,339-312,348, doi:10.1002/2017GL074837, 2017.

Stephens, B. B., Brailsford, G. W., Gomez, A. J., Riedel, K., Mikaloff Fletcher, S. E., Nichol, S., and Manning, M.: Analysis of a 39-year continuous atmospheric $CO_2$ record from Baring Head, New Zealand, Biogeosciences, 10, 2683-2697, 10.5194/bg-10-2683-2013, 2013.